# Assortative mixing in micro-architecturally annotated brain connectomes

Vincent Bazinet ®[1], Justine Y. Hansen ®[1], Reinder Vos de Wael[1],
Boris C. Bernhardt ®[1], Martijn P. van den Heuvel ®[2] & Bratislav Misic ®[1] ✉

The wiring of the brain connects micro-architecturally diverse neuronal populations, but the conventional graph model, which encodes macroscale brain connectivity as a network of nodes and edges, abstracts away the rich biological detail of each regional node. Here, we annotate connectomes with multiple biological attributes and formally study assortative mixing in annotated connectomes. Namely, we quantify the tendency for regions to be connected based on the similarity of their micro-architectural attributes. We perform all experiments using four cortico-cortical connectome datasets from three different species, and consider a range of molecular, cellular, and laminar annotations. We show that mixing between micro-architecturally diverse neuronal populations is supported by long-distance connections and find that the arrangement of connections with respect to biological annotations is associated to patterns of regional functional specialization. By bridging scales of cortical organization, from microscale attributes to macroscale connectivity, this work lays the foundation for next-generation annotated connectomics.

The brain is a complex network of anatomically connected and functionally interacting neuronal populations[1]. Representing the brain as a graph of grey matter nodes interconnected by white matter edges allows us to articulate and quantify its organizational principles. A compact set of hallmark features has been documented across organisms, spatial scales and reconstruction technologies[2]. These include communities of densely interconnected brain regions and disproportionately well connected hubs[3,4]. Together, these features promote a balance between specialization and integration[5].

An important limitation of the graph model of the brain is the assumption that all regions are the same. Yet, regions differ in their intrinsic micro-architectural attributes[6–10]. These attributes include gene expression[11–16], cellular morphology[17,18] and density[19], cell type[20], neurotransmitter receptor profiles[21–24], laminar differentiation[25–28], and myelination[29]. Understanding how the heterogeneous micro-architectural attributes of regional nodes are related to their connectional fingerprint is a fundamental question in systems neuroscience[10,30–32].

Multiple studies have shown that the arrangement of connections and regional attributes are related. For example, regions with more macroscale connections tend to have more dendritic spines, larger dendritic trees and greater neural density[17]. Moreover, regions with similar attributes are more likely to be connected with each other[14,22,27,29,33,34], suggesting a tendency for homophilic attachment. However, the assessment of the relationship between connectivity and micro-architecture is complicated by the background influence of the brain's spatial embedding on both, whereby spatially proximal regions are likely to have similar micro-architecture, but also to share anatomical connections[35–38]. Disentangling the relationships between neural wiring, regional heterogeneity and spatial embedding is a core challenge[39,40]. Furthermore, studies are often limited to a constrained set of attributes in a single organism, precluding discovery of universal principles of cortico-cortical organization.

Here we apply principled methods from network science to construct annotated connectomes. We use connectomes reconstructed from tract-tracing in model organisms as well as high-resolution in vivo

[1]McConnell Brain Imaging Centre, Montréal Neurological Institute, McGill University, Montréal, Canada. [2]Center for Neurogenomics and Cognitive Research, Vrije Universiteit Amsterdam, Amsterdam, Netherlands. ✉e-mail: bratislav.misic@mcgill.ca

imaging in humans, and annotate them with multiple micro-architectural attributes including gene expression, neuron density, receptor fingerprints and intracortical myelin. We then systematically quantify the assortativity of these annotated connectomes: the tendency of regions with similar attributes to connect with one another. In particular, we implement null models to assess the contribution of spatial constraints. We find a tendency for regions with similar annotations to connect with each other, and highlight the role of long-distance projections in connecting micro-architecturally diverse regions. We also generalize the concept of assortative mixing to address two biologically relevant questions about wiring principles of brain networks. First, we consider heterophilic assortativity: are regions enriched with one attribute more likely to be connected with regions enriched with another attribute? Second, we consider local assortativity: how similar is a region to its connected neighbors in terms of its annotations?

## Results

The results are organized as follows. We first use the assortativity coefficient to explore the relationship between connectivity and the regional distribution of nodal attributes. We then specifically look at the assortative mixing of long-range connections. Finally, we uncover heterophilic patterns of connectivity between different micro-architectural properties and extend the general concept of assortativity to the local level.

We use four different connectomes, namely a human diffusion-weighted MRI structural connectome, a human resting-state functional MRI connectome, a macaque tract-tracing connectome and a mouse tract-tracing connectome (Fig. 1a). Each connectome is annotated with micro-architectural annotations. In other words, each node in the connectome is given a local annotation score associated with a micro-architectural attribute. The human connectomes are annotated with

cortical thickness, T1w/T2w ratio (a proxy for intra-cortical myelin[41]), the ratio of excitatory-to-inhibitory neurotransmitter receptors in a region (E/I ratio), the density of neurotransmitter receptors in a region and the principal axis of gene expression (gene PC1). The macaque is annotated with cortical thickness, T1w/T2w ratio and neuron density while the mouse connectome is annotated with its principal axis of gene expression (Fig. 1b).

### Assortativity of cortical attributes

We first explore the relationship between micro-architectural annotations and connectome organization using the assortativity coefficient. For a given annotated network, assortativity is defined as the Pearson correlation between the local annotation scores of connected nodes[42]. In other words, it quantifies the tendency for nodes with similar annotation scores to be connected (Fig. 2a). An important challenge for measuring assortativity is that cortical attributes are spatially autocorrelated, and at the same time, connections also tend to form between brain regions that are proximal in space[37]. As a result, assortativity may be trivially confounded by spatial embedding. To assess how cortical attributes are related to brain connectivity, we control for this spatial autocorrelation[38,43,44]. Namely, we compare empirical assortativity coefficients to the assortativity coefficients of null annotations with preserved spatial autocorrelation (Fig. 2b).

We find that all annotations are positively assortative (i.e., brain regions tend to be connected to other regions with similar attributes), but that surrogate annotations also have positive assortativity scores (Fig. 3a). To account for the influence of spatial autocorrelation on assortativity, we compute the standardized assortativity score (z-assortativity) of each attribute relative to the null distributions of spatial autocorrelation-preserving surrogates and evaluate the significance of the scores using a two-sided permutation test, corrected

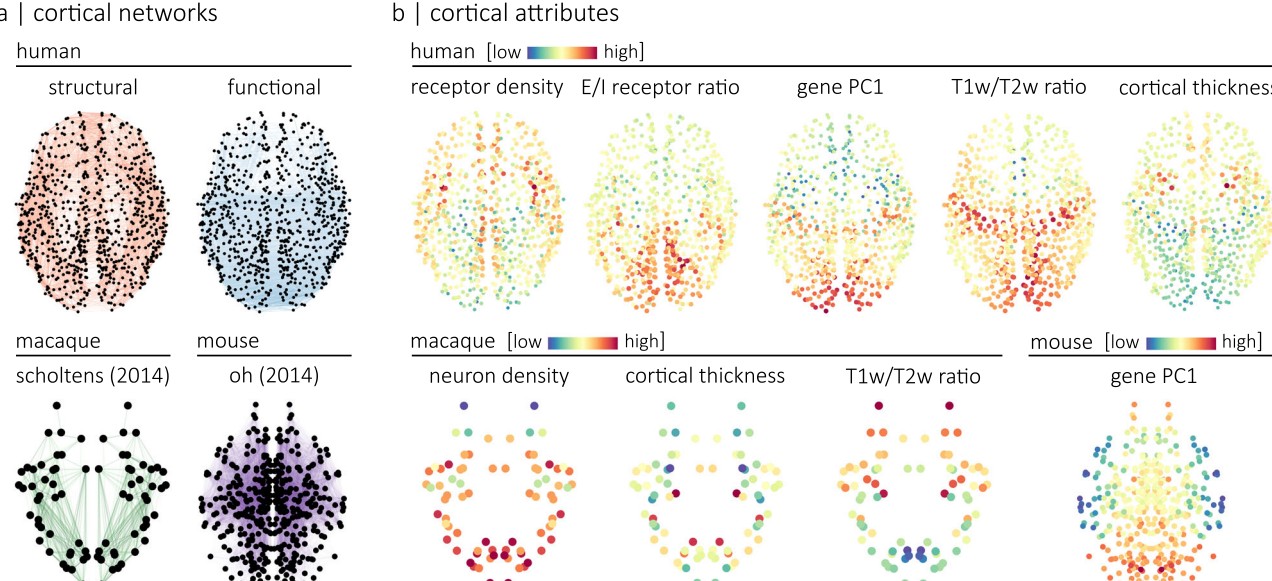

**Fig. 1 | Annotated connectomes.** We annotated four connectomes with micro-architectural attributes. **a** Connectomes include a human structural and a human functional connectome reconstructed using data from the HCP[86], a macaque connectome generated using data from the CoCoMac database and initially introduced in Scholtens et al.[17], and a mouse connectome reconstructed using data from the Allen Mouse Brain Connectivity Atlas and introduced in Oh et al.[102]. The human connectomes are parcellated according to the 800-nodes Schaefer functional atlas[89], the macaque connectome is parcellated according to a hybrid Walker-von Bonin and Bailey atlas[101] and the mouse connectome comprises 213 brain regions from the Allen Mouse Brain Atlas[103]. Spatial coordinates used for visualization of the

human and macaque connectomes correspond to the parcel centroids of their respective atlas. The spatial coordinates used for visualization of the mouse connectome were obtained from the Allen Reference Atlas, version 2 (2011). **b** Human connectomes are annotated with measures of neurotransmitter receptor density, the ratio of excitatory-to-inhibitory neurotransmitter receptors, the principal axis of gene expression (gene PC1), T1w/T2w ratio and cortical thickness. The macaque connectome is annotated with neuron density (neuron-to-cell ratio), cortical thickness and T1w/T2w ratio. The mouse connectome is annotated with the principal axis of gene expression (gene PC1).

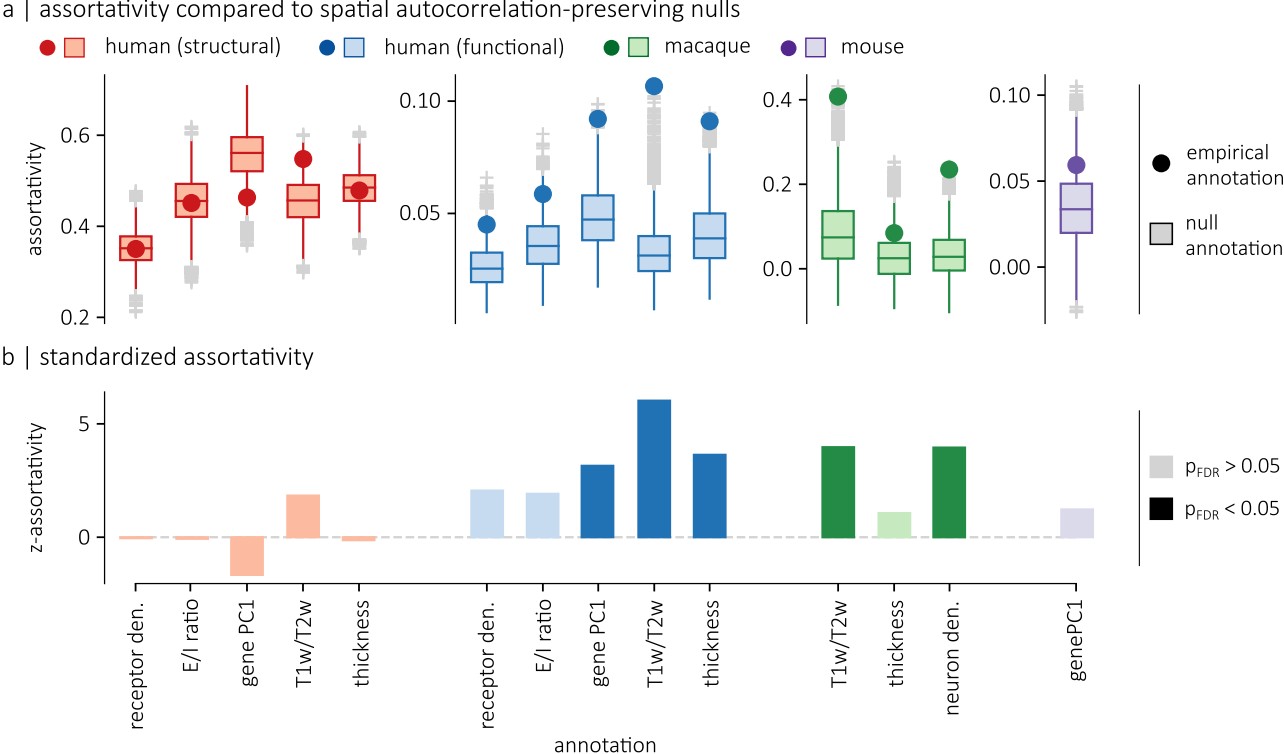

**Fig. 2 | Assortative mixing. a** Given an annotated network where each node has a local annotation score, we can quantify the tendency for nodes with similar scores to be connected using the assortativity coefficient. This coefficient is defined as the Pearson correlation between the scores of connected nodes[42]. This relationship between the scores of connected nodes can be visualized with a scatterplot of a network's edges where the position of each edge is determined by the annotation scores of its two endpoints. Here, the intersection of the two dashed lines indicates the position of the edge highlighted in the zoomed-in frame of the network. In this

example, the assortativity coefficient ($r$) is equal to 0.54. **b** To control for spatial constraints, the assortativity coefficient of an empirical annotation is compared to the assortativity coefficients of $n = 10,000$ null annotations that preserve the spatial autocorrelation of the empirical one[38,43,44]. The boxplots in **b** represent the 1st, 2nd (median) and 3rd quartiles of the null distribution; whiskers represent endpoints of the distribution. The spatial coordinates used for the visualization of the connectome corresponds to the parcel centroids of the 800-nodes Schaefer functional atlas[89].

**Fig. 3 | Standardized assortativity of micro-architectural annotations.**
**a** Assortativity of empirical annotations (points) are compared to $n = 10,000$ null annotations with preserved spatial autocorrelation (boxplots). For the human connectomes, the nulls were generated using a spatial permutation model. For the mouse and macaque connectomes, the nulls were generated using a parameterized null model. **b** Standardized assortativity scores (z-assortativity), computed relative to the spatial autocorrelation-preserving null annotations. Significance is evaluated

using a two-sided permutation test, corrected for false discovery rate (FDR). Gene PC1 ($p_{spin} = 0.0049$), T1w/T2w ratio ($p_{spin} = 0.0001$) and cortical thickness ($p_{spin} = 0.0017$) are significantly assortative on the functional connectome, while T1w/T2w ($p_{moran} = 0.002$) and neuron density ($p_{moran} = 0.0001$) are significantly assortative on the macaque connectome. Boxplots in **a** represent the 1st, 2nd (median) and 3rd quartiles; whiskers represent the non-outlier endpoints of the distribution; and + symbols represent outliers.

for false discovery rate (Fig. 3b). This method ensures that the p-values are not inflated as a result of spatial autocorrelation[38]. See Supplementary Fig. 1 for examples of p-values obtained with spatially-naive null models. We find that annotations are not significantly assortative on the human structural connectome, while gene PC1 (z-assort = 3.15, $p_{spin} = 0.0049$), T1w/T2w (z-assort = 6.02, $p_{spin} = 0.0001$) and cortical thickness (z-assort = 3.63, $p_{spin} = 0.0017$) are significantly assortative

on the human functional connectome. In the macaque connectome, we observe a significant difference between the assortativity of T1w/T2w and null annotations (z-assort = 3.98, $p_{moran} = 0.002$) as well as between neuron density and null annotations (z-assort = 3.93, $p_{moran} = 0.0001$). No significant difference is observed for the cortical thickness. In the mouse connectome, no significant difference is observed for gene PC1.

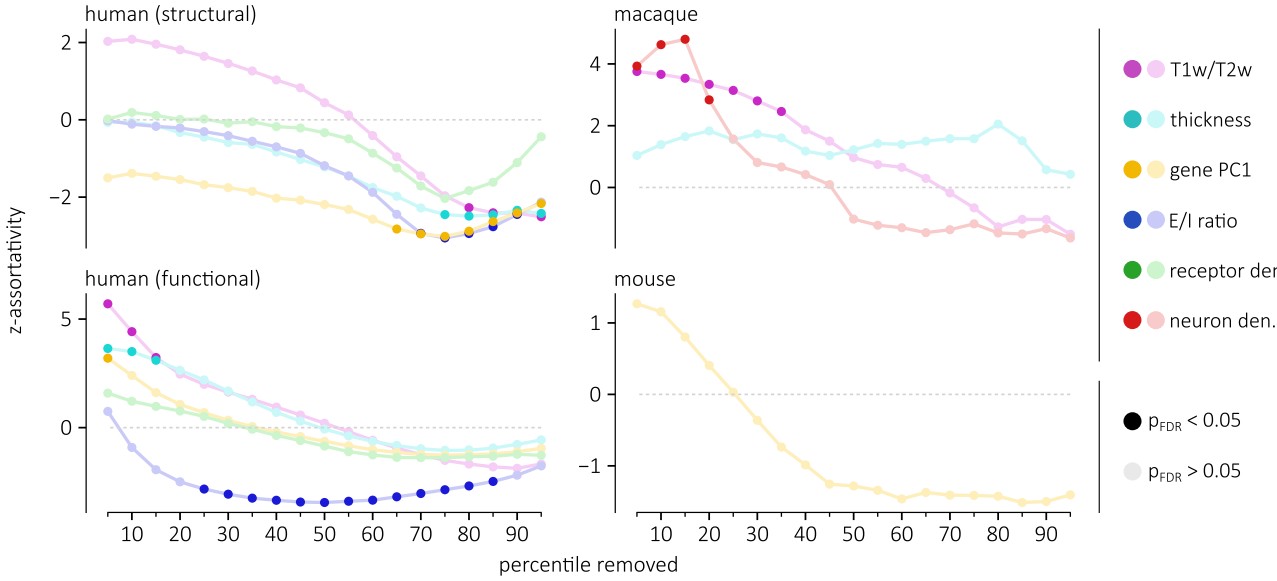

**Fig. 4 | Assortative mixing of long-range connections.** Assortativity is computed in each of the four connectomes, thresholded such that a percentile of the shortest connections are removed. These assortativity scores are standardized with respect to a null distribution of n=10,000 spatial autocorrelation-preserving nulls. Standardized assortativity scores (z-assortativity) for each annotation are displayed as a function of the percentile of connections removed in the network. For all four connectomes, annotations become less assortative as short-range connections are removed.

To ensure that the results are not sensitive to processing choices, we replicated the experiments using different parcellation schemes, single-hemisphere connectomes, an independently acquired dataset, additional spatially-autocorrelation preserving nulls models and a rank-based assortativity measure (Supplementary Figs. 2–4). Furthermore, for each assortativity results, we regressed out the potential contribution of the other annotations, confirming that the results are not confounded by relationships between pairs of annotations (Supplementary Fig. 5). We also explored the relationship between annotations using multiple linear regression and dominance analysis (Supplementary Fig. 6).

The results show that there are numerous instances where annotations that are prima facie assortative are actually not significantly assortative when we account for spatial autocorrelation. We do find instances, however, where assortativity is significantly larger than expected from the brain's spatial embedding and, interestingly, these findings are consistent with recent reports in the literature. The significant standardized assortativity of neuron density in the macaque cortex is consistent with reports that neuron density is more related to the existence of connections than geodesic distance[45]. Significant assortativity in the functional connectome is also consistent with recent reports that functional connectivity gradients are closely aligned with multiple micro-architectural properties[6,13,28,29].

### Geometric contributions to assortativity
Recent theories suggest that long-distance connections in the structural connectome enhance the diversity of a brain region's inputs and outputs[46]. Long-distance connections may thus potentially promote communication between regions with dissimilar attributes. This idea, however, has never been formally tested from the perspective of biological annotations.

We therefore explored how the standardized assortativity of different attributes, relative to null annotations with preserved spatial auto-correlation, varies as we consider connections of different lengths. For all four connectomes and for each annotation, we compute the standardized assortativity across thresholded connectomes where a given percentile of the shortest connections is removed. We find that as short-distance connections are removed − leaving behind

the longest connections − the standardized assortativity of all annotations across all four connectomes decreases (Fig. 4). Notably, with 80% of the human structural connectome's connections removed, four annotations become significantly disassortative (two-sided permutation test, FDR-corrected): E/I ratio (z-assort = − 2.94, $p_{spin}$ = 0.021), gene PC1 (z-assort = − 2.88, $p_{spin}$ = 0.021), T1w/T2w ratio (z-assort = − 2.27, $p_{spin}$ = 0.0499) and cortical thickness (z-assort = − 2.49, $p_{spin}$ = 0.039). In other words, the remaining long-range connections link regions with attributes that are more dissimilar than we would expect from the brain's spatial embedding. Again, these results are consistent across multiple methodological choices (Supplementary Figs. 2–4). This confirms the notion that long-distance connections increase the diversity of a region's inputs and outputs, supporting the integration of information between micro-architecturally dissimilar regions.

### Heterophilic mixing of cortical attributes
In the previous sections, we used the assortativity coefficient to ask if two areas are more likely to be connected if they are enriched with the same attribute. In other words, we quantified the homophilic mixing of micro-architectural attributes. An equally important question is whether there exists heterophilic mixing in the brain. In other words, are two regions more likely to be connected if one region is enriched with one attribute while the other is enriched with a different attribute? Cortico-cortical connectivity may indeed reflect interactions between pairs of distinct attributes. For instance, it has been hypothesised that the noradrenergic and cholinergic systems influence in distinct ways large-scale dynamical processes in the brain[47]. Laminar organization also appears to be closely related to brain connectivity[48–50]. We next ask if the heterogeneous distribution of pairs of attributes from multi-member classes of annotation − neurotransmitter receptor profiles and laminar differentiation − is reflected in the connectivity of the brain.

To address these questions, we analyze two datasets (Fig. 5a). The first is a positron emission tomography (PET)-derived atlas of 18 receptors and transporters from 9 neurotransmitter systems[22]. The second is the thickness of individual cortical layers in the Merker-stained BigBrain histological atlas[26,51]. To quantify heterophilic mixing,

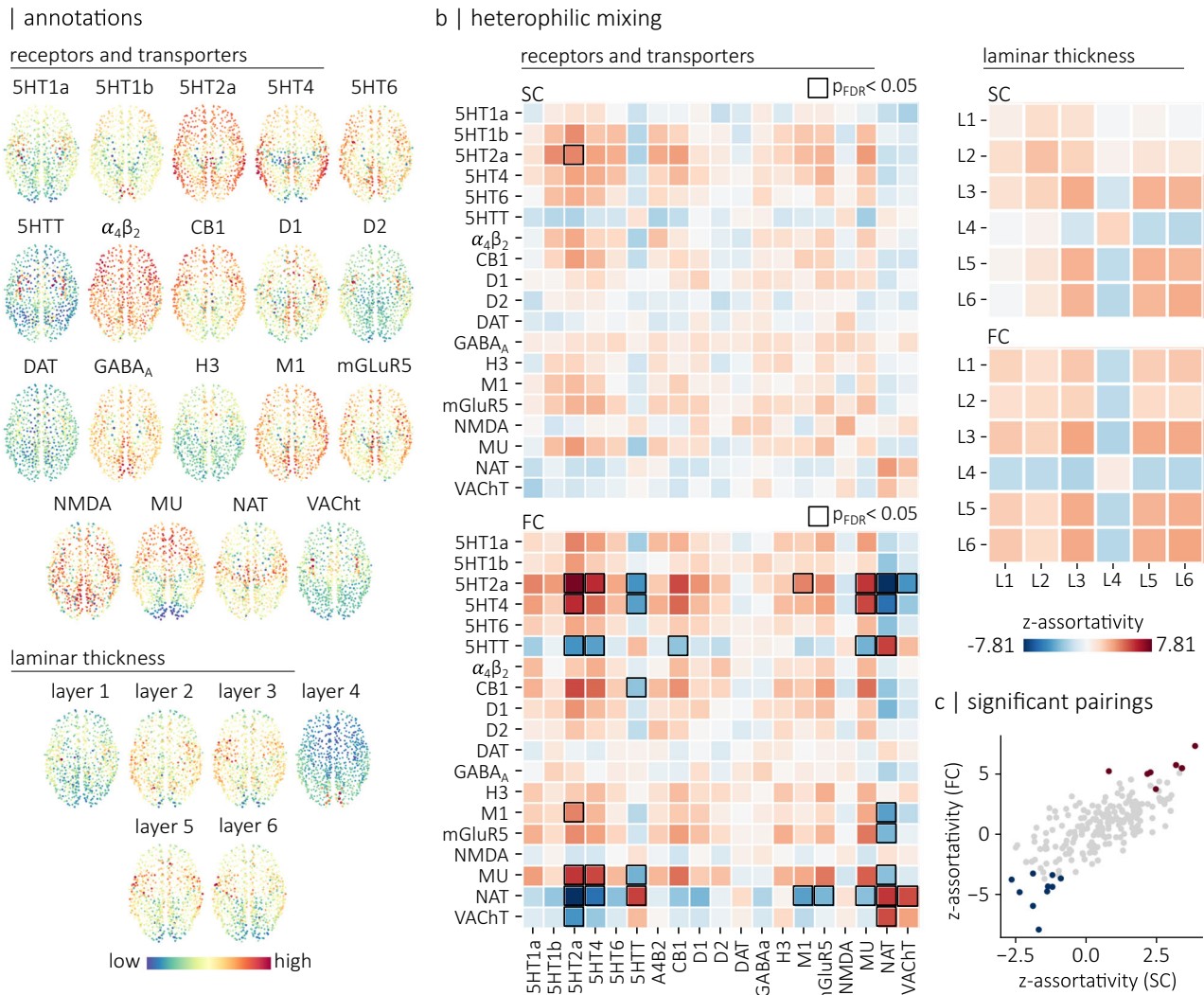

**Fig. 5 | Heterophilic mixing. a** Topographic distribution of PET-derived brain maps showing the density of 18 transporters and receptors[22], as well as the topographic distribution of laminar thicknesses extracted from the Merker-stained BigBrain histological atlas[26,51]. The spatial coordinates corresponds to the parcel centroids of the 800-nodes Schaefer functional atlas[89] (**b**) Heterophilic mixing matrices for the receptors/transporter annotations (left) and for the laminar thickness annotations (right). Positive values indicate that regions that score highly on an annotation **x** are more likely to be connected to regions that score highly on an annotation **y** than would be expected from the brain's spatial embedding. Negative values indicate that regions that score high on an annotation **x** are more likely to be connected to regions that score low on an annotation **y**. Black squares highlight statistically significant relationships ($p < 0.05$), evaluated using a two-sided permutation test with spatial autocorrelation-preserving null annotations and corrected for multiple comparisons (FDR). **c** Relationship between the standardized assortativity of annotation pairs in the structural connectome (SC) and in the functional connectome (FC). We find a strong relationship between z-assortativity in SC and z-assortativity in FC ($r = 0.74$). Highlighted in red are annotation pairs that are significantly assortative in FC. Highlighted in blue are annotation pairs that are significantly disassortative in FC.

we extend the concept of assortativity to pairs of annotations. In other words, we compute the assortativity coefficient for pairs of annotations such that the annotation at endpoint $i$ represents an attribute **x** and the annotation at endpoint $j$ represents a different attribute **y**. Figure 5b shows the heterophilic mixing matrices of the structural and functional connectomes for both receptor density and laminar thickness. The assortativity results are standardized with respect to spatial autocorrelation-preserving null annotations generated by permuting the attributes on the surface of the brain (spins). Importantly, these permutations preserve the correlation between brain maps. By controlling for both the brain's spatial embedding and the correlation between brain maps, the analyses specifically assess the relationship between brain connectivity and the heterogeneous distributions of pairs of micro-architectural attributes. Positive values in each matrix indicate that a region that scores highly on an annotation **x** is more likely to be connected to a region that scores highly on an annotation **y**

than expected from the spatial distribution of each annotation. Negative values indicate that a region that scores high on an annotation **x** is more likely to be connected to a region that scores low on an annotation **y** than expected from the spatial distribution of each annotation.

Several salient associations emerge that are consistent with prior intuitions and qualitative descriptions in the literature (Fig. 5b). For the laminar thickness, we find consistent mixing patterns for both the structural and functional connectomes. Namely, layers III, V and VI are assortative with respect to each other, but disassortative with respect to layer IV. In other words, we find that brain areas with a prominent layer IV tend to preferentially connect with brain areas with thin layers III, V and VI, whereas brain regions with prominent layers III, V and VI tend to preferentially connect with each other. Interestingly, in the functional connectome, these mixing patterns can be explained by the relationship between each laminar thickness map and the functional

hierarchy, defined as the main axis of variance in the brain's functional connectivity matrix[6,52]. Indeed, assortative mixing between two annotations arises when attributes have positive or negative relationships with the main axis of brain connectivity, while disassortative mixing arises when attributes have opposite relationships with this axis (Supplementary Fig. 7ab). For the patterns of assortativity observed between layer IV and layers III, V and VI, it has been shown that the thickness of layer IV is most prominent in the primary visual cortex (ref. 26), while the thicknesses of layers III, V and VI increases along the visual processing hierarchy (ref. 26). Similarly, we find here that the thickness of layer IV is positively correlated with the functional hierarchy while the thicknesses of layers III, V and VI are negatively correlated with the functional hierarchy (Supplementary Fig. 7c). These opposing relationships with the functional hierarchy, such that a prominent layer IV is associated with unimodal brain regions while prominent layer III, V and VI are associated with multimodal regions explain the mixing patterns observed (Supplementary Fig. 7c).

This general idea also extends to receptors where we broadly find evidence of disassortative mixing for pairs of receptors and transporters predominantly expressed in brain regions on opposite ends of this functional hierarchy. More specifically, we find 10 pairs of receptors and transporters that are significantly disassortative (Supplementary Table 1). A clear pattern emerges, whereby each pair involves a transporter (5HTT, NAT or VAChT) and a receptor. This is in line with their relationship with the functional hierarchy: transporter maps tend to be anticorrelated with the unimodal-transmodal hierarchy, while the identified receptor maps tend to be positively correlated with the unimodal-transmodal hierarchy (Supplementary Fig. 7d). Altogether, these results highlight a network-mediated balance between transporter and receptor density.

We also find a significant assortative relationship in the functional connectome between vesicular acetylcholine transporters (VaChT) and NAT (z-assort = 5.10, $p_{spin} = 0.028$). Transporters are generally expressed pre-synaptically. Thus, the results show that regions densely innervated by cholinergic neurons tend to be connected with regions densely innervated by noradrenaline neurons, above and beyond what would be expected from the brain's spatial embedding. Our findings therefore support the idea that these two systems interact with each other and with the brain's topology to influence large-scale dynamical processes[47]. Assortative relationships tend to be similar in both structural and functional connectomes (Fig. 5c). They are also replicable with alternate parcellations and datasets (Supplementary Fig. 8). Collectively, these complex heterophilic mixing patterns show evidence of how macroscale white matter projections support interfacing among neuronal populations with diverse microscale attributes.

## Local assortative mixing

In the previous two sections, we explored how cortical attributes align with the underlying connectome at the global level. Here we extend this concept to the local level and consider the extent to which individual regions connect to regions with similar attributes. We first compute the absolute difference between the local annotation scores of connected nodes (edge differences; Fig. 6a, left). To quantify the local assortativity of a region, we then compute the average of its edge differences to its connected neighbours, weighted by the connection weight between the two nodes (mean difference; Fig. 6a, right). This local assortativity score represents how different a region is from other regions it is anatomically connected with in terms of its biological attributes.

Importantly, annotation scores that deviate from the mean of the distribution are on average more dissimilar to any other score (Fig. 6b). To account for this, we define the homophilic ratio of a node as the ratio between its mean difference with connected neighbors and its mean difference with all the nodes in the network (Fig. 6b). Nodes that have large homophilic ratios are nodes that tend to connect to brain regions with more dissimilar properties (disassortative) while nodes that have small homophilic ratios tend to connect to regions with more similar properties (assortative).

The homophilic ratios of all five annotations on the structural connectome are shown in Fig. 6c. The homophilic ratios are consistent across parcellations and datasets (Supplementary Fig. 9). We also computed the homophilic ratio of each annotation on the functional connectome (Supplementary Fig. 10). For the structural connectome, we summarized the assortativity of each node by computing their averaged homophilic ratio across all five annotations and quantified the relationship between homophilic ratio and mean connection distance as well as node strength (Fig. 6d). We find a significant relationship for both mean connection distance ($r = 0.35$, $p_{spin} = 0.0001$), and node strength ($r = 0.21$, $p_{spin} = 0.021$). In other words, disassortative regions have, on average, longer connections, which is consistent with our previous findings that long distance connections tend to be disassortative (Fig. 4). Also, the results highlight a general trend in the structural connectome where brain regions that have large node strength (i.e., "hub" regions) tend to be more disassortative. In other words, the hub regions of the brain tend to connect to regions that have dissimilar micro-architectural attributes.

We next explored whether these findings are consistent across communities of the brain. We clustered the human structural connectomes into 9 communities of highly interconnected regions and computed the mean homophilic ratio and mean node strength inside each community. We find that the relationships between node strength, mean connection distance and homophilic ratio hold for all communities, with the exception of the dorsolateral prefrontal community (Supplementary Fig. 11). Brain regions in this community have, on average, the largest node strengths but the smallest homophilic ratios. In other words, contrary to the other hub regions of the brain, nodes in DLPFC tend to preferentially connect to regions that have similar micro-architectural attributes.

Finally, we ask whether the homophilic ratio of a brain region is related to its functional specialization. We extracted brain maps of probabilistic associations between functional keywords and individual voxels using the Neurosynth meta-analytic engine[53] and correlated the brain maps associated with 123 cognitive and behavioural terms[54] with the averaged homophilic ratio brain map. We then grouped the correlation scores based on the cognitive category associated with each term (Fig. 6e). We find, on average, significant (two-sided permutation test, FDR-corrected) negative correlations between the average homophilic ratio brain map and both the "action" ($r = -0.20$, $p = 0.02$) and the "Executive/Cognitive control" ($r = -0.21$, $p = 0.0001$) categories. In other words, regions with large activation during tasks of these categories tend to have small homophilic ratios. Conversely, we find that cognitive terms associated to the "learning" and "memory and emotion" categories have, on average, activation maps that are positively correlated with the average homophilic ratio brain map (learning and memory: $r = 0.11$, $p = 0.003$; emotion: $r = 0.15$, $p = 0.003$). In other words, regions with large activation during tasks of these categories tend to have large homophilic ratios. This suggests that executive functions are subtended by a network of areas that tend to connect to other areas with similar microscale attributes. Conversely, integrative functions such as consolidation and memory are subtended by a set of medial temporal structures that project to regions with diverse microscale attributes. Correlations for individual cognitive terms are shown in Supplementary Fig. 12.

## Discussion

In the present report we investigate the link between connectome architecture and microscale biological annotations. More specifically, we ask whether brain regions with similar attributes are more likely to be connected with each other above and beyond the role of spatial proximity. We systematically assess the tendency for global and local

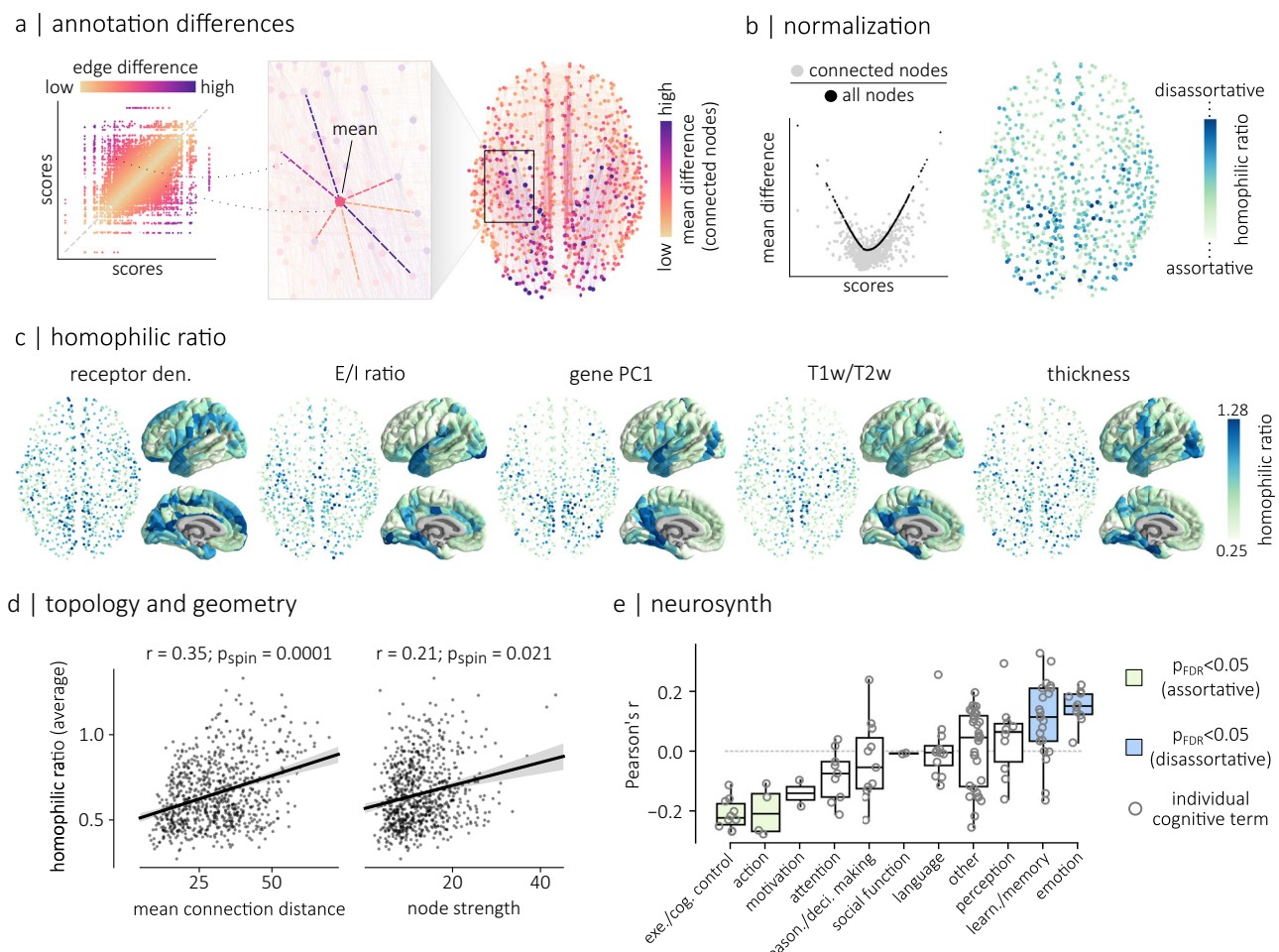

**Fig. 6 | Local assortative mixing. a** We first computed the weighted average of the absolute differences between each brain region's local annotation score and the local annotation scores (mean differences) of its directly-connected neighbors. **b** The mean difference between a node's annotation score and the annotation scores of all the other nodes in the network (black points in the scatterplot) is directly related to its annotation score. A similar relationship exists when considering connected neighbors (gray points in the scatterplot). Therefore, a node's homophilic ratio is defined as its mean difference with connected neighbors, divided by its mean difference with all the nodes in the network. **c** Homophilic ratios of five micro-architectural attributes, for the human structural connectome. **d** The homophilic ratio, averaged across the five micro-architectural attributes shown in **c**, is significantly correlated (permutation tests against $n = 10,000$ spatial autocorrelation-preserving nulls, two-sided) with the mean connection distance (right; $r = 0.35$, $p_{spin} = 0.0001$, CI = [0.29, 0.41]) and the strength (left; $r = 0.21$, $p_{spin} = 0.02$, CI = [0.15, 0.28]) of a node. Regression lines are shown for both relationships. Shaded bands represent the 95% confidence intervals of the regression

estimates. **e** We correlated 123 parcellated brain maps of probabilistic associations between functional keywords and individual voxel activation[53] with the averaged homophilic ratio of each node. These keywords are separated into 11 different cognitive categories: executive/cognitive control, action, motivation, attention, reasoning and decision making, social function, language, other, perception, learning and memory, and emotion. On average, we find significant (two-sided permutation test, FDR-corrected) negative correlations between the averaged homophilic ratio brain map and the activation maps associated to the "action" ($r = -0.20$, $p = 0.02$) and the "Executive/Cognitive control" ($r = -0.21$, $p = 0.0001$) categories. Conversely, the averaged homophilic ratio brain map is positively correlated with activation maps associated to the "learning and memory" ($r = 0.11$, $p = 0.003$) and "emotion" ($r = 0.15$, $p = 0.003$) categories. Boxplots in **e** represent the distribution's 1st, 2nd (median) and 3rd quartiles; whiskers represent the distribution's endpoints. Spatial coordinates in **a**–**c** correspond to the parcel centroids of the 800-nodes Schaefer functional atlas[89]. Cortical surfaces in **c** are visualized on the pial surface of the FreeSurfer[145] fsaverage template using PySurfer[142].

homophilic mixing across a variety of attributes. We show that mixing between micro-architecturally diverse neuronal populations is supported by long-distance connections. Finally, we highlight how brain connectivity supports heterophilic mixing patterns between neurotransmitter systems and cortical layers.

The present work builds upon an exciting new direction in network neuroscience to jointly consider macroscale networks and microscale attributes[8,17,18,31,39]. Contemporary theories emphasize the link between cytoarchitectonic similarity and synaptic connectivity[48–50,55,56]. Numerous recent reports related macroscale connectivity to microscale annotations, including gene expression[14,57–60], cytoarchitecture[27–29,33,45] and neurotransmitter receptor profiles[22,24]. While some of these studies have shown evidence of global assortative mixing for specific attributes

and connectomes types, the extent to which these findings can be generalized to wiring principles across network reconstruction techniques, species, spatial scales and annotations remains unknown. By considering a broad range of annotation in multiple connectome datasets, the present work comprehensively studies how connectivity between neural populations depends on their micro-architecture.

Importantly, the spatial embedding of the brain constrains its organization[37]. Microscale attributes are graded across the cortex[6,10,61], which means that most cortical attributes are spatially autocorrelated[13,38,44,62]. There is also a concomitant prevalence of short-distance connections compared to long-distance connections[63], with connection probability and strength typically decaying exponentially with spatial distance[35,36,64–66]. Collectively, these principles of cortical

organization necessitate careful consideration and methodological control of spatial effects when studying the relationship between connectivity and annotations[13,38,43,44,62,67].

Here, we rigorously assess how network wiring, micro-architectural features and spatial embedding are intertwined from the perspective of assortativity. Although degree assortativity – whether nodes with similar degrees are more likely to be connected with each other – has previously been studied in brain networks[68–72], the fundamental idea is more general and can be applied to any nodal features. In this sense, assortativity, combined with spatially-constrained null models, is the ideal framework to study connectome annotations.

For the functional connectome, we find a significant relationship between connectivity and gene PC1, T1w/T2w, and cortical thickness. This is consistent with previous reports that these three properties are related to the principal gradient of functional connectivity[6,13,29], which can be thought of as the dominant pattern of assortativity in functional connectomes[52]. For the structural connectome, we find that assortative connectivity heavily depends on the annotation itself. While we find significant assortativity in the macaque for T1w/T2w and neuron density (consistent with previous reports[45]), we find also numerous counter-examples of annotations that are not assortative beyond the background effect of spatial embedding. For instance, we find that receptor density and E/I ratio, T1w/T2w ratio, cortical thickness and gene PC1 are not assortative in the human connectome, nor is gene PC1 assortative on the mouse connectome. Collectively, these results support a general tendency for cytoarchitectonically similar regions to be connected, but also highlight the fact that not all features conform to this wiring principle.

By definition, assortativity means that brain regions will be connected to regions that are similar to themselves; a functional consequence is that regions are less likely to be exposed to diverse inputs. Importantly, we find that long distance connections are an architectural feature that potentially serves to diversify inputs to a brain region. Indeed, the longest connections, in structural (diffusion and tract-tracing) and functional connectomes are significantly disassortative, meaning that they are more likely to connect dissimilar regions than expected from the brain's spatial embedding. This is in line with the notion that greater prevalence of short-range connections[35,36,64–66], which presumably entail lower material and metabolic cost[63], is counter-balanced by a small number of high-cost, high-benefit long-range connections that support communication between regions with diverse functions[73,74]. Previous studies have found that long-range connections, which are heterogeneously distributed along microarchitectural and cognitive hierarchies[75,76], help to shorten communication pathways[4], and bridge specialized modules[46]. Our results build on this literature by showing that long-range connections are also more likely to be placed between regions that are biologically distinct.

We also extend the conventional framework of assortativity to ask two biologically important questions about heterophilic and local homophilic mixing. The notion of heterophilic mixing becomes particularly convenient when we study a multi-member class of annotations, and wish to know whether a node enriched with one attribute is likely to be connected to a node enriched with another attribute. In the brain, two notable examples are receptor profiles and laminar differentiation, both of which have been associated to patterns of synaptic connectivity[22,49]. For instance, ascending cholinergic and noradrenergic neuromodulatory systems are thought to provide complementary mechanisms to balance segregation (cholinergic) and integration (noradrenergic)[47]. Our results highlight a tendency for cortical areas that are rich in cholinergic and noradrenergic transporters to be connected, offering a potential anatomical mechanism to maintain this balance. We also find that the thickness of granular layer IV, which is more prominent in sensory regions, is disassortative with

the thickness of the other layers of the cortex. This is in line with previous findings that have shown that sensory regions, such as visual cortex, form segregated modules in macroscale structural and functional networks[68,77–79]. How heterophilic mixing is organized between different classes (e.g., areas enriched with specific layers connected to areas enriched with specific receptors[80]) remains an exciting question that could be readily addressed with the present framework.

Finally, we zoom in to specific regions and assess the extent to which their local biological annotations conform to the annotations of their connected neighbours, thereby generalizing the concept of global assortativity to the local level. Using meta-analytic decoding, we find that regions that connect to biologically similar regions tend to be associated with executive function. This may reflect the fact that these areas (e.g., dorsolateral prefrontal cortex), which are cytoarchitecturally distinct from other prefrontal regions[81], form a highly interconnected module in the structural connectome[68]. Conversely, regions that connect with biologically dissimilar regions tend to be associated with memory function. This may reflect the idea that these regions (e.g., medial temporal cortex) are involved in integrating signals from multiple specialized circuits[71]. Collectively, these results show that the arrangement of connectivity patterns with respect to biological annotations may ultimately shape patterns of regional functional specialization.

The present results should be interpreted with respect to important methodological limitations. First, human structural connectomes were reconstructed using diffusion imaging, a technique that is known to yield multiple false positives and false negatives[82,83], and which cannot be used to infer directionality. Although we replicated the results using high fidelity tract-tracing and histology in multiple animal models, further development in reconstructing human white-matter connectomes is needed. Second, despite the fact that we tried to be as extensive and comprehensive as possible in our choice of annotations, spanning molecular, cellular and laminar attributes, the final set of annotations is incomplete. Exciting technological and data-sharing advances will eventually permit even more detailed and comprehensive biological annotations to be studied using this framework. Third, the number and the resolution of samples across annotations varies. To mitigate the impact of sampling and resolution differences on our results, we parcellated each brain map according to the same atlas, which effectively maps them to a set number of brain regions uniformly distributed across the cortex and across datasets. Relationship between connectivity and attributes depend on how brain regions are defined (i.e., parcellations). We systematically studied multiple parcellations, but how best to delineate functional territories of the cortex remains an open challenge in the field[84,85].

In summary, the present work bridges scales of cortical organization, from microscale attributes to macroscale connectivity. By carefully controlling the background effect of spatial embedding, we systematically assess how connectivity is interdigitated with a broad range of micro-architectural attributes and empirically test multiple theories about the wiring of cortical networks. This work lays the foundation for next-generation annotated connectomics.

## Methods
### Connectomes

**Human connectomes (HCP).** The human connectomes were generated using data from the Human Connectome Project S900 release[86]. Scans from $N = 327$ unrelated participants (age $28.6 \pm 3.73$ years, 55% females) were used to reconstruct a consensus structural and functional connectome. Informed consent was obtained for all subjects (the protocol was approved by the Washington University Institutional Review Board as part of the HCP). The participants were scanned in the HCP's custom Siemens 3T "Connectome Skyra" scanner, and the acquisition protocol included a high angular resolution imaging

(HARDI) sequence and four resting state fMRI sessions. Briefly, the dMRI data was acquired with a spin-echo EPI sequence (TR = 5520 ms; TE = 89.5 ms; FOV = 210 × 180 mm²; voxel size = 1.25 mm³; b-value = three different shells i.e., 1000, 2000, and 3000 s/mm²; number of diffusion directions = 270; and number of b0 images = 18) and the resting-state fMRI data was acquired using a gradient-echo EPI sequence (TR = 720 ms; TE = 33.1 ms; FOV = 208 × 180 mm²; voxel size = 2 mm³; number of slices = 72; and number of volumes = 1200). Additional information regarding the acquisition protocol is available at ref. 86.

The data was pre-processed according to the HCP minimal pre-processing pipelines[87] and structural connectomes were reconstructed from the dMRI data using the MRtrix3 package[88]. Grey matter was parcellated into 800 cortical regions according to the Schaefer functional atlas[89] and fiber orientation distributions were generated using a multi-shell multi-tissue constrained spherical deconvolution algorithm[90,91]. The initial tractogram was generated with 40 million streamlines, with a maximum tract length of 250 and a fractional anisotropy cutoff of 0.06. Spherical-deconvolution informed filtering of tractograms (SIFT2) was used to reconstruct whole brain streamlines weighted by cross-section multipliers[92]. More information regarding the individual network reconstructions is available at ref. 93.

A group consensus structural network was then built such that the mean density and edge length distribution observed across individual participants was preserved[94]. The weights of the edges in the consensus networks correspond to the log-transform of the number of streamlines in the parcels, averaged across participants for whom these edges existed. A group-average functional connectivity matrix was constructed by concatenating the regional fMRI BOLD time series of all four resting-state sessions from all participants and computing the zero-lag Pearson correlation coefficient between each pair of brain regions. Experiments were also replicated using connectomes parcellated into 400 cortical regions, again according to the Schaefer functional atlas[89], and without log-transforming the edge weights.

**Human connectomes (Lausanne).** Our experiments were also replicated in a second dataset collected at the Lausanne University Hospital (N = 67; age 28.8 ± 9.1 years, 40% females)[95]. Participants were scanned in a 3-Tesla MRI Scanner (Trio, Siemens Medical, Germany). Informed consent was obtained for all subjects (the protocol was approved by the Ethics Committee of Clinical Research of the Faculty of Biology and Medicine, University of Lausanne, Switzerland). Details regarding data acquisition, pre-processing and network reconstruction are available at ref. 95. Briefly, the data acquisition protocol included a magnetization-prepared rapid acquisition gradient echo (MPRAGE) sequence (1mm in-plane resolution, 1.2 mm slice thickness), a diffusion spectrum imaging (DSI) sequence (128 diffusion-weighted volumes and a single b0 volume, maximum b-value 8000 s/mm², 2.2 × 2.2 × 3.0 mm voxel size), and a gradient echo-planar imaging (EPI) sequence sensitive to blood-oxygen-level-dependent (BOLD) contrast (3.3 mm in-plane resolution and slice thickness with a 0.3 mm gap, TR 1920 ms, resulting in 280 images per participant). Grey matter was parcellated into either 219 and 1000 equally sized parcels[96]. The Connectome Mapper Toolkit was used for the initial signal processing[97] while gray and white matter were segmented from the MPRAGE volume using freesurfer[98]. Structural connectivity matrices were reconstructed for individual participants using deterministic streamline tractography on reconstructed DSI data. 32 streamline propagations were initiated per diffusion direction and per white matter voxel.

Again, a group consensus structural network was built such that the mean density and edge length distribution observed across individual participants was preserved[94]. The weights of the edges correspond to the log-transform of the streamline densities, averaged across participants and scaled to values between 0 and 1. fMRI volumes

were corrected for physiological variables (regression of white matter, cerebrospinal fluid, as well as motion), BOLD time series were subjected to a lowpass filter and motion "scrubbing"[99] was performed. A group-average functional connectivity matrix was reconstructed using the same procedure as described above.

**Macaque connectome.** The macaque connectome was initially introduced in Scholtens et al.[17] and was generated using data from the CoCoMac database, an online repository of tract-tracing experiments[100]. The parcellation used for the network reconstruction is an hybrid between the Walker-von Bonin and Bailey atlases[101] and contains 39 non-overlapping cortical regions. The network was constructed such that a connection is assigned to pairs of brain regions if (i) a tract is reported in a least five studies in the database and ii) at least 66% of the reports are positive. The connectome is directed and each edge is weighted between 1 and 3 based on the averaged reported strength of the connection.

**Mouse connectome.** The mouse connectome was generated by Oh et al.[102] using data from the Allen Mouse Brain Connectivity Atlas. This connectivity atlas contains high-resolution images acquired from 469 injection experiments performed in the right hemisphere of C57BL/6J male mice. Each experiment produced 140 high-resolution (0.35 μm) coronal sections of EGFP-labelled axonal projections which were then registered to the Allen Mouse Brain Atlas[103]. A weighted directed connectome of 213 brain regions was constructed. The strength of each connection was obtained by fitting a linear connectivity model to the data. The connectivity data, the name of the 213 brain regions as well as the euclidean distance between each region was obtained from the supplemental material of Oh et al.[102]. The spatial coordinates used for visualization were obtained from the Allen Mouse Reference Atlas, version 2 (2011).

## Annotations

**Human annotations.** Cortical thickness and T1w/T2w ratio were extracted from high-resolution structural scans made available by the Human Connectome Project[86]. For the HCP connectomes, the morphometric measures were obtained for each one of the 201 individuals used to reconstruct the connectomes and averaged, for each node of the parcellations, across subjects. For the Lausanne connectomes, the morphometric measures, averaged across subjects of the S1200 release, were fetched and parcellated using neuromaps[61].

The principal axis of transcriptional variation across the human cortex (gene PC1) captures a hierarchy of transcriptomic specialization across the human cortex[6,13] and has been used in multiple studies (refs. 11, 34, 104–106. It was computed using the Allen Human Brain Atlas (AHBA; https://human.brain-map.org/)[11], which provides regional microarray expression data from six post-mortem brains (1 female, ages 24–57, 42.5 ± 13.38). The AHBA data was pre-processed and mapped to the parcellated brain regions using the abagen toolbox (https://github.com/rmarkello/abagen)[107]. During pre-processing, the MNI coordinates of tissue samples were updated to those generated via non-linear alignment to the ICBM152 template anatomy (https://github.com/chrisgorgo/alleninf). Microarray probe information was re-annotated for all genes using data provided by Arnatkeviciute and colleagues[108]. For bilateral connectomes, microarray expression samples were mirrored across hemispheres to increase spatial coverage. Then, probes were filtered by only retaining those that have a proportion of signal to noise ratio greater than 0.5. When multiple probes indexed the expression of the same gene, the one with the most consistent pattern of regional variation across donors was selected. Samples were then assigned to individual regions in the parcellations. If a sample was not found directly within a parcel, the nearest sample, up to a 2mm-distance, was selected. If no samples were found within 2 mm of the parcel, the sample closest to the centroid of the empty

parcel across all donors was selected. To reduce the potential for misassignment, sample-to-region matching was constrained by hemisphere and gross structural divisions (i.e., cortex, subcortex/brainstem, and cerebellum, such that e.g., a sample in the left cortex could only be assigned to an atlas parcel in the left cortex). All tissue samples not assigned to a brain region in the provided atlas were discarded. Tissue sample expression scores were then normalized across genes using a scaled robust sigmoid function[14], and were rescaled to a unit interval. Expression scores were also normalized across tissue samples using the same procedure. Microarray samples belonging to the same regions were then aggregated by computing the mean expression across samples for individual parcels, for each donor. Regional expression profiles were finally averaged across donors to obtain a single genes × brain regions matrix. From the 15,632 genes listed in this matrix, 1906 brain-specific genes were used to compute the principal axis of transcriptional variation using Principal component analysis. The list of brain-specific genes was obtained from[13]. The principal axis obtained (gene PC1) captures 20% of the total gene expression variance in the matrix.

Receptor density information was collected for 18 different neurotransmitter receptors and transporters from a total of 25 different studies as described in ref. 22. The receptor maps include 5HT1a[109], 5HT1b[109,110], 5HT2a[111], 5HT4[111], 5HT6[112], 5HTT[111], $\alpha_4\beta_2$[113], CB1[114], D1[115], D2[116], DAT[117], GABA$_A$[118], H3[119], M1[120], mGluR5[121,122], NMDA[123,124], MU[125], NAT[126], VAChT[127,128]. Positron emission tomography (PET) images registered to the MNI space were parcellated and receptors/transporters with more than one mean image of the same tracer (5-HT1b, D2, VAChT) were combined using a weighted average. Tracer maps, each corresponding to a single receptor/transporter where then normalized across regions to values between 0 and 1. Receptor density was computed as the average density, across all 18 receptors while an excitatory/inhibitory ratio was computed as the ratio between the mean density of excitatory receptors and the mean density of inhibitory receptors. Excitatory receptors include: 5HT2a, 5HT4, 5HT6, $\alpha_4\beta_2$, D1, M1, mGluR5. Inhibitory receptors include: 5HT1a, 5HT1b, CB1, D2, GABA$_A$, H3, MU.

Laminar thickness information was extracted from the Merkel-stained BigBrain histological atlas[26,51]. Individual cortical layers were individually segmented with a convolutional neural network, as described in Wagstyl et al.[26], and the laminar surfaces were made available on the BigBrain Project website (https://ftp.bigbrainproject.org/). Laminar thickness was computed as the Euclidean distance between each pair of corresponding vertices on each 3D surfaces. The data was then parcellated to the Schaefer (800 nodes) parcellation[89] using surface parcellation files in the BigBrain space that are also available on the BigBrain Project website.

**Macaque annotations.** Three macaque annotations were obtained. The cortical thickness and T1w/T2w ratio cortical maps are originally from Donahue et al.[129] and were extracted from the structural MRI scans of 19 adult macaques (T1w and T2w, 0.5 mm isotropic). These brain maps were publicly shared in the BALSA database[130] (https://balsa.wustl.edu/study/show/W336). The cortical maps were first parcellated using a 91 regions parcellation scheme (M132). The data was then further parcellated to the WBB atlas using a region-wise mapping provided in[131]. Neuron density information was extracted from brain tissues of an Old World macaque monkey[19] and mapped to the WBB atlas using a mapping provided in[17].

**Mouse annotations.** The principal axis of gene expression variation (gene PC1) was computed using data from the Allen Mouse Brain Atlas[103]. This atlas contains gene expression profiles, obtained using in-situ hybridization, from more than 20,000 genes. Expression density within each of the 213 structures defined in the oh2014 connectome was computed by combining/unionizing grid voxels with the same 3-D

structural label. The data was obtained using the mouse module of the abagen toolbox (https://github.com/rmarkello/abagen)[107]. To facilitate comparison between genes, we normalized expression levels for each gene. We then computed the principal axis of gene expression variation across brain regions using principal component analysis.

**Spatial autocorrelation-preserving null annotations**
We controlled for the brain's spatial constraints using null models that preserve the spatial autocorrelation of the empirical attributes. The use of spatial permutation nulls (spin nulls) was prioritized since they tend to be more conservative[38]. These nulls require the use of spherical projections of the brain, which were not available in animal datasets. For the animal datasets, we therefore relied on a parameterized null model that uses Moran spectral randomization (Moran nulls)[132]. All the results were also replicated with a third null model originally proposed by Burt and colleagues (Burt nulls), and the results obtained with spin nulls (i.e. for the human connectomes) were also replicated with the Moran nulls. The spin, Moran and Burt nulls were respectively implemented with the neuromaps (https://github.com/netneurolab/neuromaps)[61], brainspace (https://github.com/MICA-MNI/BrainSpace)[132] and brainSMASH (https://github.com/murraylab/brainsmash)[44] toolboxes. Using these annotations, we performed two-sided permutation tests. In other words, a distribution of 10,000 null annotations was generated and a p-value was computed by comparing the empirical result to the distribution of null results obtained with the null annotations. P-values were also corrected for False discovery rate (FDR) using the Benjamini-Yekutieli procedure, which controls for false discovery rate under arbitrary dependence assumptions[133].

**Spin nulls.** The original framework for this spatial permutation model was introduced in Alexander-Bloch et al.[43] and consists in generating null distributions by applying random rotations to spherical projections of the brain. Here, we use a framework adapted to parcellated data originally proposed in Vázquez-Rodríguez et al.[134]. Namely, we select for each parcel the vertex closest to its center of mass on the spherical projection of the fsaverage surface. We then apply a rotation to the coordinates of these centers of mass and reassign to each parcel the value of the closest rotated parcel. To preserve homotopy across hemispheres, the rotations are generated independently for one hemisphere and then mirrored across the anterior-posterior axis for the other.

**Moran nulls.** The generation of spatially constrained nulls using Moran spectral randomization was first proposed in the ecology literature[135] and relies on a spatially-informed weight matrix **W**. The eigenvectors of **W** provide an estimate of the autocorrelation in the brain and are used to impose a similar spatial structure on random, normally distributed surrogate data. Here, **W** is defined as the inverse of the distance matrix between brain regions. For the human connectomes, the distance between pairs of parcels was defined as the mean geodesic distance between every vertex pair in both parcels. In the animal connectomes, it was defined as the euclidean distance between both parcels. Data was generated separately for each hemisphere using the same random seed to obtain null annotations that preserve homotopy across hemispheres.

**Burt nulls.** This parameterized null model was originally proposed in Burt et al.[44]. First, the empirical brain map is randomly permuted. Then, this permuted brain map is spatially smoothed and re-scaled to re-introduce the spatial autocorrelation (SA) of the empirical brain map. The smoothing process is achieved via the following transformation:

$$\mathbf{y} = |\beta|^{1/2}\mathbf{x} + |\alpha|^{1/2}\mathbf{z}, \tag{1}$$

where **y** is the surrogate map, **x** is the permuted data and **z** is a vector of random gaussian noise. The $\alpha$ and $\beta$ parameters are estimated via a least-square optimization between variograms of the original and permuted data. By maximizing the fit between the variograms of the original and permuted data, we ensure that the SA of the surrogate map matches the SA of the empirical map. Again, the distances between pairs of parcels in the human connectomes were obtained by averaging the geodesic distance between every vertex in the two parcels while they were obtained by computing the euclidean distance between each parcel in the animal connectomes. Also, data was generated separately for each hemisphere using the same random seed to obtain null annotations that preserve homotopy across hemispheres. The hyper-parameters used were the default parameters provided by the brainSMASH software[44] (https://github.com/murraylab/brainsmash).

## Assortativity

To study the relationship between distributions of cortical attributes and the topological architecture of our connectomes, we relied on the assortativity coefficient, which is defined as the Pearson correlation between the annotations of connected nodes[42]. More precisely, given an adjacency matrix **A**, where $a_{ij}$ represents the strength of the connection between brain regions $i$ and $j$, and a vector of annotations **x**, where $x_i$ represents the annotation attributed to node $i$, the assortativity of a network, with respect to **x** is defined as:

$$r_{\mathbf{x}} = \sum_{ij} \frac{a_{ij}}{2m} \tilde{x}_i \tilde{x}_j, \tag{2}$$

where $2m$ corresponds to the sum of the edge weights in the network and $\tilde{x}_i$ represents the standardized score of the annotation attributed to node $i$:

$$\tilde{x}_i = \frac{x_i - \bar{\mathbf{x}}}{\sigma_{\mathbf{x}}}, \tag{3}$$

$\bar{\mathbf{x}}$ corresponds to the expected value of **x** and $\sigma_{\mathbf{x}}$ corresponds to the standard deviation of **x**:

$$\bar{\mathbf{x}} = \frac{1}{2m} \sum_i k_i x_i \tag{4}$$

$$\sigma_{\mathbf{x}} = \sqrt{\frac{1}{2m} \sum_i k_i (x_i - \bar{\mathbf{x}})^2}. \tag{5}$$

$k_i$ corresponds to the strength of node $i$.

## Ranked-based assortativity

More generally, given an annotation x, we can define the vectors $\mathbf{x}^{(i)}$ and $\mathbf{x}^{(j)}$ as the annotations of endpoints $i$ and endpoints $j$ across all edges $(i,j)$ in the network. In other words, each entry in the vectors $\mathbf{x}^{(i)}$ and $\mathbf{x}^{(j)}$ represent the annotations of nodes connected by an edge in the network. The assortativity coefficient can then be defined as a weighted Pearson correlation between these two vectors, weighted by the weight of the connection between each edge.

By ranking the annotation scores in the vectors $\mathbf{x}^{(i)}$ and $\mathbf{x}^{(j)}$, we can compute a rank-based assortativity coefficient that corresponds to the weighted Spearman correlation between these two vectors. This rank-based coefficient allows us to evaluate the existence of monotonic relationships between the annotations of connected brain regions.

## Partial assortativity

To evaluate the partial assortativity of an annotation **y** with respect to **x**, another vector of annotation score, we fit a linear regression

between the scores $\mathbf{y}^{(i)}$ and $\mathbf{x}^{(i)}$, as well as between the scores $\mathbf{y}^{(j)}$ and $\mathbf{x}^{(i)}$, for all edges $(i,j)$ in the network, and then compute the weighted Pearson correlation between the residuals of these regressions.

## Multiple linear regression and dominance analysis

To characterize the assortativity of a brain region across multiple annotations, we built multiple linear regression models. These models were used to explain the annotation score of a brain region for a specific annotation from the annotation scores, across all annotations available, of its connected brain regions. Formally, given a vector of annotation scores $\mathbf{y}^{(j)}$ representing the annotations score of endpoints $j$ for all edges $(i,j)$ and vectors of annotation scores for independent variables $\mathbf{x}_1^{(i)}, \mathbf{x}_2^{(i)}, ..., \mathbf{x}_n^{(i)}$, which represent the annotations scores of endpoints $i$, for variables $\mathbf{x}_1, \mathbf{x}_2, ..., \mathbf{x}_n$, we modeled the annotation score of the dependent variable **y** from the annotation scores of the independent variables $\mathbf{x}_1, \mathbf{x}_2, ..., \mathbf{x}_n$ using a linear regression model:

$$\mathbf{y}^{(j)} = b_0 + b_1 \mathbf{x}_1^{(i)} + b_2 \mathbf{x}_2^{(i)} + \cdots + b_n \mathbf{x}_n^{(i)}. \tag{6}$$

where $b_1, b_2, ..., b_n$ are the regression coefficients and $b_0$ is the intercept. The regression coefficients were estimated using weighted least squares with the strength of the connections between pairs of brain regions used as weights.

We then estimated the contribution of each annotation to those linear regression models using dominance analysis. This technique consists in fitting the same regression model using every combination of input variables[136] and then evaluating the total dominance of a variable, defined as the average of the relative increase in R2 observed when adding this variable to each submodel. This measure indicates how each regressor variable contributes to the model while accounting for interactions between regressors.

## Heterophilic mixing

The assortativity coefficient can also measure the heterophilic mixing between pairs of annotations. We define heterophilic mixing as the tendency for nodes with a given standardized scores for an attribute **x** to connect to nodes with similar standardized scores for another attribute **y**. The assortativity coefficient, for pairs of annotations **x** and **y** is defined as:

$$r_{\mathbf{x},\mathbf{y}} = \sum_{ij} \frac{a_{ij}}{2m} \tilde{x}_i \tilde{y}_j, \tag{7}$$

## Homophilic ratio

To quantify the extent to which individual regions connect to regions with similar attributes, we computed the homophilic ratio of each node. This measure is a ratio between the weighted average of the absolute difference of a node's annotation with its neighbors and the averaged absolute difference of this node's annotation with all the other nodes in the network. More precisely, the homophilic ratio $h$ of a given node $i$ for an annotation **x** is defined as

$$h_{\mathbf{x}}(i) = \frac{\sum_j \frac{a_{ij}}{k_i} |x_i - x_j|}{\frac{1}{n} \sum_j |x_i - x_j|}, \tag{8}$$

where $n$ is the number of nodes in the connectome.

## Mean connection distance

The mean homophilic ratio of each node was compared to its mean connection distance. This measure is defined as the average distance between a node and its connected neighbors, weighted by the weight of each connection. More precisely, the mean connection distance

(MCD) of a node $i$ is defined as

$$MCD(i) = \frac{1}{2m}\sum_j d_{ij}a_{ij}, \quad (9)$$

where $d_{ij}$ corresponds to the Euclidean distance between nodes $i$ and $j$.

## Community detection

Communities are groups of nodes with dense connectivity among each other. The Louvain method was used to identify a community assignment or partition that maximizes the quality function $Q$[137]:

$$Q = \frac{1}{2m}\sum_{i,j}\left[A_{ij} - \gamma\frac{s_is_j}{2m}\right]\delta(c_i.c_j), \quad (10)$$

where $A_{ij}$ is the weight of connection between nodes $i$ and $j$, $s_i$ and $s_j$ are the directed strengths of $i$ and $j$, $m$ is a normalizing constant, $c_i$ is the community assignment of node $i$ and the Kronecker $\delta$-function $\delta(u,v)$ is defined as 1 if $u=v$ and 0 otherwise. The resolution parameter $\gamma$ scales the importance of the null model and effectively controls the size of the detected communities: larger communities are more likely to be detected when $\gamma < 1$ and smaller communities (with fewer nodes in each community) are more likely to be detected when $\gamma > 1$.

To detect stable community assignments for the structural connectome we initiated the algorithm 100 times at each value of the resolution parameter and consensus clustering was used to identify the most representative partitions[138]. This procedure was repeated for a range of 100 resolutions between $\gamma = 0.25$ and $\gamma = 7.5$. We then quantified the similarity between pairs of consensus partitions using the $z$ score of the Rand index[139]. Ultimately, we chose the 9-communities consensus partition obtained at $\gamma = 1.23$ because the generated partitions obtained for this value of $\gamma$ showed high mutual similarity and persisted through stretches of $\gamma$ values. The whole procedure was implemented using code available in the netneurotools python toolbox (https://github.com/netneurolab/netneurotools).

## Probabilistic activation maps

Using the Neurosynth meta-analytic engine[53] we extracted brain maps of probabilistic associations between functional key words and individual voxels, synthesized from results from more than 15,000 published fMRI studies. The probabilistic measures quantify the probability that a given term is reported in a study and that there is activation observed in a given voxel. It can be interpreted as a quantitative representation of how regional fluctuations in activity are related to psychological processes. We analyzed the functional maps associated to 123 cognitive and behavioural terms from the Cognitive Atlas (ref. 54, ranging from umbrella terms ("attention", "emotion") to specific cognitive processes ("visual attention", "episodic memory"), behaviours ("eating", "sleep"), and emotional states ("fear", "anxiety").

These cognitive terms were grouped into 11 cognitive categories. These categories consist of "Action", "Learning and Memory", "Emotion", "Attention", "Reasoning and Decision Making", "Executive/Cognitive control", "Social Function", "Perception", "Motivation", "Language" and "other". Lists of terms associated with each category can be found here: http://www.cognitiveatlas.org/concepts/categories/all. To evaluate whether, on average, terms associated to one of these categories are significantly correlated with the average homophilic ratio brain map, we performed a two-sided permutation test where we randomly permuted the categories associated with each term and computed the average correlations obtained from these permuted categories. This was repeated 10000 times, and a p-value was computed by comparing the average correlations obtained with the empirical categories with the null distribution of average correlations obtained with the permuted categories. The p-values obtained

were then corrected for false discovery rate using the Benjamini-Yekutieli procedure[133].

## Reporting summary

Further information on research design is available in the Nature Portfolio Reporting Summary linked to this article.

## Data availability

The data used to conduct the analyses is available at https://github.com/netneurolab/bazinet_assortativity. More generally, the HCP dataset[86] is available at https://db.humanconnectome.org/data/projects/HCP_1200, the Lausanne dataset is available at https://doi.org/10.5281/zenodo.2872624, the Allen Human Brain Atlas[11] is available at https://human.brain-map.org, the receptor density atlas[22] is available through neuromaps (https://github.com/netneurolab/neuromaps)[61], the Allen Mouse Brain Connectivity Atlas[102] is available at https://connectivity.brain-map.org, the Allen Mouse Brain Atlas[103] is available at https://mouse.brain-map.org/static/atlas, the CoCoMac database[100] is available at http://cocomac.g-node.org, the macaque neuron density[19] data is available at https://doi.org/10.1073/pnas.1010356107, the macaque structural MRI scans[129] are publicly available in the BALSA database[130] (https://balsa.wustl.edu/study/show/W336) and the BigBrain data[26,51] is available at https://ftp.bigbrainproject.org/. Source data are provided with this paper.

## Code availability

The code used to conduct the analyses and generate the figures presented in this paper is available at https://github.com/netneurolab/bazinet_assortativity and directly relies on the following open source Python packages: BrainSMASH[44], BrainSpace[132], Matplotlib[140], neuromaps[61], NumPy[141], PySurfer[142], Scipy[143] and Seaborn[144].

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

## Acknowledgements

We thank Laura Suarez, Golia Shafiei, Bertha Vazquez-Rodriguez, Ross Markello, Zhen-Qi Liu, Filip Milisav, and Andrea Luppi for insightful comments. VB acknowledges support from the Fonds de Recherche Québec - Nature et Technologie and from the Natural Sciences and Engineering Research Council of Canada (NSERC). BM acknowledges support from the NSERC, Canadian Institutes of Health Research (CIHR), Brain Canada Foundation Future Leaders Fund, the Canada Research Chairs Program, the Michael J. Fox Foundation, and the Healthy Brains for Healthy Lives initiative.

## Author contributions

Conceptualization: V.B. and B.M.; Methodology: V.B. and B.M.; Formal Analysis: V.B.; Data Curation: V.B., J.Y.H., R.V., B.C.B.; Writing - Original Draft: V.B., B.M.; Writing - Review & Editing: J.Y.H., R.V., B.C.B., M.P.v.d.H.; Visualization: V.B.; Supervision: B.M.

## Competing interests

The authors declare no competing interests.
