## [Peer Review File · Nature Communications]

Assortative mixing in micro-architecturally annotated brain connectomesREVIEWER COMMENTS

Reviewer #1 (Remarks to the Author):

Bazinet et al, address an interesting question: the relationship between the connectivity of anatomical regions (human functional and white matter structural; monkey and mouse structural) and their similarities and differences with respect to a variety of local morphological, histological, cytochemical, and genetic expression properties. Though the idea is attractive, the realisation raises some serious methodological questions that complicate the interpretation of the results.

1. Wide variation in annotation sampling density and resolution appears to be left completely unmodelled. There is only one BigBrain, histamine receptor data is derived from only 8 patients, and transcription data is both sparsely sampled and derived from a small number of brains. It is difficult to see how comparisons across annotations can be confidently made when there is so much unmodelled variation in noise and coverage. Note that since the sampling of many of the annotations is not random, this is also a source of bias.

2. Each annotation is examined either in isolation or paired with one other. Ignoring similarities and differences distributed across multiple characteristics means the association of any single annotation may be confounded by another. In any event, if the objective is to characterise the correspondence between connectivity and regional similarities then it makes much more sense to characterise each region across multiple dimensions taken together.

3. Transcription patterns are reduced to the first principal component of >15,000 gene profiles. We are not told what proportion of the variance that captures, but it is likely to be too small to serve as a faithful representation of the underlying signal.

4. Everything rests on a purely linear univariate measure of regional similarity based on Pearson correlation. It is very likely that the diverse annotations studied here exhibit all sorts of non-linear variations, both across the brain and in their associations with each other. For example, the mutual information between two regions may be much higher than a simple linear index suggests. If systematic differences in non-linearity across annotations exist, then the comparison becomes confounded.

5. What are referred to as spatial nulls are not null models under the assumption of no structure but rather models of local similarity. It is not clear why or how local similarity should be discounted. It is not accidental that the brain exhibits local smoothness and if the objective is to understand variations with distance there is no logic in arbitrarily downweighting proximity, especially when spatial homogeneity

may vary substantially across the brain. The authors use several such presumed nulls, varying the choice across datasets without giving a clear reason.

6. Though we are not given the number of tests the authors clearly perform many, yet there is no mention of any correction for multiple comparisons. This is not trivial where correlation between tests is likely, as in the regional comparisons. In any event, without it there is little we can conclude from the p values cited.

Reviewer #2 (Remarks to the Author):

The authors add an important analysis to the current studies of brain connectomes – they investigate how regional biological attributes affect white matter connectivity, thus linking traditional neuroanatomic approaches with current brain connectivity studies. Specifically, the authors study assortative mixing in annotated connectomes, hypothesizing that regions with similar vs. dissimilar micro-architectonic attributes (such as molecular, cellular and laminar annotations, including gene expression, neurotransmitter receptors, neuron density, laminar thickness and intracortical myelin) show distinct connectivity patterns. The analyses are based on four connectome datasets from three species, i.e. humans, macaques and mice and the authors use a novel null model to assess the contribution of spatial constraints (i.e. to control for spatial autocorrelation). They observe that regions with similar annotations tend to connect with each other and that micro-architecturally diverse regions are connected via long-distance projections.

These are very interesting data and analyses. My main questions are regarding the interpretation of results.

The results regarding heterophilic mixing of brain areas with a prominent layer 4 are particularly interesting. The authors write that “layer IV is most prominent in the primary visual cortex, while the thicknesses of layers III, V and VI increase along the sensory processing hierarchy” Can this fully explain the strong effects observed? Are the authors referring to visual sensory processing, or sensory processing in general (I assume the former)? The authors could (rather easily?) test their hypothesis of a gradual change of heterophilic mixing of primary visual cortex along the visual processing hierarchy.

The authors write that they „find evidence of disassortative mixing for pairs of receptors that are predominantly expressed in brain regions on opposite ends of the processing hierarchy“ - could they explain this in more detail, i.e. what hierarchy and how this was defined and what receptor subtypes are

involved? Indeed, most receptors and their subtypes are involved in a multitude of very different functions.

How were receptors selected for these analyses? There are 6 different subtypes of serotonin receptors, but no ionotropic glutamate receptors and only one metabotropic glutamate receptor; specifically, why are NMDAR and AMPA receptors missing, given that glutamate is the main excitatory neurotransmitter?

I am also not fully convinced yet about the interpretation of how the “homophilic ratio of a brain region shapes its functional specialization”. The homophilic ratio was positively correlated with memory performance, and it appears plausible to interpret that e.g. the medial temporal lobe is “involved in integrating signals from multiple specialized circuits”. However, the homophilic ratio was negative correlated with executive function – but executive functions specifically also require integration of signals from very different brain regions and circuits, and they are not only subserved by a highly interconnected module centering around the DLPFC.

It would also be good to see the correlation coefficients for all 123 cognitive and behavioural terms to get a better idea about the distribution of effects (might be best in the suppl. Material)

Dear Dr. Walters and Reviewers,

Thank you for the constructive feedback. Following your comments and suggestions, we have thoroughly revised the manuscript. In this letter, we respond to each of the reviewer's comments in detail. Reviewer comments are in **bold font** and our responses are in regular font. New text in the revised manuscript is in light blue.

Major changes include:

- (1) Additional sensitivity analyses to test for interactions and potential confounds among annotations.
- (2) Correction for false discovery rate throughout the manuscript.
- (3) Updated the human receptor dataset to include the ionotropic NMDA receptor.
- (4) Clarified the rationale for using specific autocorrelation-preserving null models.
- (5) Clarified how patterns of assortativity and disassortativity are subtended by the unimodal-transmodal hierarchy.

Reviewer #1:

Bazinet et al, address an interesting question: the relationship between the connectivity of anatomical regions (human functional and white matter structural; monkey and mouse structural) and their similarities and differences with respect to a variety of local morphological, histological, cytochemical, and genetic expression properties. Though the idea is attractive, the realisation raises some serious methodological questions that complicate the interpretation of the results.

1. Wide variation in annotation sampling density and resolution appears to be left completely unmodelled. There is only one BigBrain, histamine receptor data is derived from only 8 patients, and transcription data is both sparsely sampled and derived from a small number of brains. It is difficult to see how comparisons across annotations can be confidently made when there is so much unmodelled variation in noise and coverage. Note that since the sampling of many of the annotations is not random, this is also a source of bias.

We agree with the Reviewer that there are variations in the sampling density of the annotations and in the size of our samples.

We however used multiple strategies to mitigate these issues. First, all brain maps were parcellated to ensure that the annotations are uniformly sampled across the cortex, and therefore comparable across datasets. Second, the datasets used in this study are the most spatially comprehensive available and have been extensively validated. For instance, the Allen Human Brain atlas has been validated via in-situ hybridization (Hawrylycz et al., 2012). The Big Brain is itself constructed using Merker-staining histology, a gold-standard for the study of human cytoarchitecture, and the receptor data has been validated against gold-standard autoradiography (Hansen et al., 2022). Third, these datasets are commonly used in the literature, which facilitates comparison with other reports (e.g. Allen Human Brain Atlas: Burt et al., 2018; Gao et al., 2020; Shafiei et al., 2020; Deco et al., 2021; BigBrain data: Wei et al., 2018; Receptor data: Hansen et al., 2022). Fourth, we validated the results in independent samples and replicated our results in animal models. Notably, to validate the results obtained with the human transcriptional data, we used mouse gene expression profiles obtained using in-situ hybridization, for more than 20,000 genes.

Furthermore, the coverage of all human annotations is complete and uniform across the cortex, with the exception of one: the human transcriptional data. Importantly, however, this dataset was processed according to state of the art recommendations that involve removing genes with expression values that did not exceed a threshold associated with background expression, as well as removing genes with variable expression across the different brains (Arnatkeviciute et al., 2019, Markello et al., 2020). Also, we do not consider transcriptional data for individual genes, but instead study the dominant axis of gene transcription, a gradient commonly used in neuroimaging studies. This reduces the likelihood of bias due to poor signal associated with specific single genes.

We now address these limitations in our discussion section.

(“Discussion” section, paragraph #9):

“Second, despite the fact that we tried to be as extensive and comprehensive as possible in our choice of annotations, spanning molecular, cellular and laminar attributes, the final set of annotations is incomplete. Exciting technological and data-sharing advances will eventually permit even more detailed and comprehensive biological annotations to be studied using this framework. Third, the number and the resolution of samples across annotations varies. To mitigate the impact of sampling and resolution differences on our results, we parcellated each brain map according to the same atlas, which effectively map them to a set number of brain regions uniformly distributed across the cortex and across datasets. Relationships between connectivity and attributes will however depend on how these brain regions are defined (i.e. parcellations). We systematically studied multiple parcellations to ensure that our results were not biased by our choice of parcellation, but how best to delineate functional territories of the cortex remains an open challenge in the field.”

Arnatkevičiūtė, A., Fulcher, B. D., & Fornito, A. (2019). A practical guide to linking brain-wide gene expression and neuroimaging data. *Neuroimage*, *189*, 353-367.

Burt, J. B., Demirtaş, M., Eckner, W. J., Navejar, N. M., Ji, J. L., Martin, W. J., ... & Murray, J. D. (2018). Hierarchy of transcriptomic specialization across human cortex captured by structural neuroimaging topography. *Nature neuroscience*, *21*(9), 1251-1259.

Deco, G., Kringelbach, M. L., Arnatkeviciute, A., Oldham, S., Sabaroedin, K., Rogasch, N. C., ... & Fornito, A. (2021). Dynamical consequences of regional heterogeneity in the brain's transcriptional landscape. *Science Advances*, *7*(29), eabf4752.

Gao, R., van den Brink, R. L., Pfeffer, T., & Voytek, B. (2020). Neuronal timescales are functionally dynamic and shaped by cortical microarchitecture. *Elife*, *9*, e61277.

Hansen, J. Y., Shafiei, G., Markello, R. D., Smart, K., Cox, S. M., Nørgaard, M., ... & Misic, B. (2022). Mapping neurotransmitter systems to the structural and functional organization of the human neocortex. *Nature Neuroscience*, 1-13.

Hawrylycz, M. J., Lein, E. S., Guillozet-Bongaarts, A. L., Shen, E. H., Ng, L., Miller, J. A., ... & Jones, A. R. (2012). An anatomically comprehensive atlas of the adult human brain transcriptome. *Nature*, *489*(7416), 391-399.

Markello, R. D., Arnatkeviciute, A., Poline, J. B., Fulcher, B. D., Fornito, A., & Misic, B. (2021). Standardizing workflows in imaging transcriptomics with the abagen toolbox. *Elife*, *10*, e72129.

Shafiei, G., Markello, R. D., De Wael, R. V., Bernhardt, B. C., Fulcher, B. D., & Misic, B. (2020). Topographic gradients of intrinsic dynamics across neocortex. *Elife*, 9, e62116.

Wei, Y., Scholtens, L. H., Turk, E., & Van Den Heuvel, M. P. (2018). Multiscale examination of cytoarchitectonic similarity and human brain connectivity. *Network Neuroscience*, 3(1), 124-137.

2. Each annotation is examined either in isolation or paired with one other. Ignoring similarities and differences distributed across multiple characteristics means the association of any single annotation may be confounded by another. In any event, if the objective is to characterise the correspondence between connectivity and regional similarities then it makes much more sense to characterise each region across multiple dimensions taken together.

We agree with the Reviewer that the assortativity scores observed for a specific annotation might be confounded by other annotations. We explored this possibility with additional experiments. For each assortativity score, we computed their partial assortativity with respect to the other annotations. Namely, we regressed-out the potential contributions of each annotation and computed weighted correlations between the residuals. By computing the significance of the correlation scores with a permutation test, we find significant correlations for all annotations in the human structural and functional connectomes. In other words, we find that the assortativity scores obtained are not confounded by other annotations. For the macaque connectome, the assortativity scores obtained for cortical thickness were not significant, which is consistent with what we found in the manuscript. However, the assortativity scores obtained for T1w/T2w ratio and neuron density were still significant after regressing out the other annotations.

We now describe these experiments in the results section, and we have added a supplementary figure (Fig. S4) that show our findings. The relevant methodology was also added to the methods section.

(“Results” section, “Assortativity of cortical attributes” subsection, paragraph #2):

“Furthermore, for each assortativity results, we regressed out the potential contribution of the other annotations, confirming that the results are not confounded by relationships between pairs of annotations (Fig. S4).

(“Methods” section, “Assortativity” subsection, “Partial assortativity” sub-subsection):

“To evaluate the partial assortativity of an annotation y with respect to x , another vector of annotation score, we fit a linear regression between the scores $y^{(i)}$ and $x^{(i)}$, as well as between the scores $y^{(j)}$ and $x^{(j)}$, for all edges (i,j) in the network, and then compute the weighted Pearson correlation between the residuals of these regressions.”

a | structural connectome

b | functional connectome

c | macaque connectome

Figure S4. Partial assortativity | To explore whether the assortativity scores observed for specific annotations are confounded by other annotations, we computed the partial assortativity of all annotations. Namely, for each annotation we regressed out the potential contributions of the other annotations and computed a weighted correlation between the residuals. Partial assortativity scores are shown for the annotations of the human structural (**a**), human functional (**b**) and macaque (**c**) connectomes. The left-most columns show the assortativity results presented in the main text. The heatmaps on the right show the partial assortativity scores, where each row consists of the partial assortativity scores of a given annotation, with the scores of each of the remaining annotations regressed out. By computing the significance of the correlation scores with a permutation test, we find significant correlations for all annotations of the human connectomes ($p_{FDR} < 0.05$). In other words, we find that the assortativity scores obtained are not confounded by other annotations. For the macaque connectome, the assortativity scores obtained for cortical thickness were not significant, which is consistent with our main findings. However, the assortativity scores obtained for T1w/T2w ratio and neuron density were still significant after regressing out the other annotations ($p_{FDR} < 0.05$).

We also agree with the Reviewer that the assortativity of a brain region could be characterized using multiple annotations simultaneously. To address this possibility, we built multiple linear regression models using all annotations, and computed their coefficient of determination (R^2). To further explore the association between each annotation, we performed a dominance analysis on each model (Budescu, 1993). The analysis allowed us to evaluate the relative contribution of each regressor in the regression models. This is done by generating a regression model using each subset of regressors, and then computing, for each regressor, the average of the relative increase in R^2 obtained when adding this regressor to each submodel.

We evaluated the significance of R^2 scores by comparing them to those obtained using 1000 permuted annotations. To preserve the dependencies between each annotation, the same permutation was used across all annotations. Annotations were permuted using spatial autocorrelation-preserving permutations models (we get R^2 scores that are significant for all annotations and for all connectomes when using spatially-naive permutations: see our reply to the Reviewer's comment #5 for an explanation on why this is the case). The significance scores were corrected for False Discovery Rate using the Benjamini-Yekutieli procedure (Benjamini & Yekutieli, 2001).

We find interesting results. In the structural connectome, linear regression models that incorporate all five annotations are better at predicting the T1w/T2w ratio of brain regions than models that incorporate permuted annotations. Particularly, cortical thickness significantly augments predictions of T1w/T2w ratio, which is in-line with studies showing that cortical thickness and T1w/T2w ratio are tightly coupled across cortex (Huntenburg et al., 2017). In the functional connectome, we obtain significant R^2 scores for the three annotations that are significantly assortative (T1w/T2w, thickness, gene PC1). Interestingly, we find that these three annotations all significantly contribute to the performance of each linear regression model, which is again consistent with the literature (Burt et al., 2018). In the macaque connectome, we also obtain significant R^2 scores for all three annotations. However, for cortical thickness, the main contributions come from T1w/T2w ratio and neuron density. Collectively, these results show that each annotation complement each other, emphasizing the benefits of integrating them in multimodal analyses.

We now describe these results in the revised manuscript and have added a supplementary figure (Fig. S5) The relevant methodology was also added to the methods section.

("Results" section, "Assortativity of cortical attributes" subsection, paragraph #2):

"We also explored the relationships between annotations using multiple linear regression models and dominance analysis (Fig. S5).

("Methods" section, "Multiple linear regression and dominance analysis" subsection):

"To characterize the assortativity of a brain region across multiple annotations, we built multiple linear regression models. These models were used to explain the annotation score of a brain region for a specific annotation from the annotation scores, across all annotations available, of its connected brain regions. Formally, given a vector of annotation scores $\mathbf{y}^{(j)}$ representing the annotations score of endpoints j for all edges (i, j) and vectors of annotation scores for independent variables $\mathbf{x}^{(i)}_1, \mathbf{x}^{(i)}_2, \dots, \mathbf{x}^{(i)}_n$, which represent the annotations scores of endpoints i , for variables $\mathbf{x}_1, \mathbf{x}_2, \dots, \mathbf{x}_n$, we modeled the annotation score of the dependent variable \mathbf{y} from the annotation scores of the independent variables $\mathbf{x}_1, \mathbf{x}_2, \dots, \mathbf{x}_n$ using a linear regression model:

$$\mathbf{y}^{(j)} = b_0 + b_1\mathbf{x}^{(i)}_1 + b_2\mathbf{x}^{(i)}_2 + \dots + b_n\mathbf{x}^{(i)}_n.$$

where b_1, b_2, \dots, b_n are the regression coefficients and b_0 is the intercept. The regression coefficients were estimated using weighted least squares with the strength of the connections between pairs of brain regions used as weights.

We then estimated the contribution of each annotation to those linear regression models using dominance analysis. This technique consists in fitting the same regression model using

every combination of input variables (Budescu, 1993) and then evaluating the total dominance of a variable, defined as the average of the relative increase in R² observed when adding this variable to each submodel. This measure indicates how each regressor variable contributes to the model while accounting for interactions between regressors.”

Figure S5. Multiple linear regression and dominance analysis | For each annotation, and for each network, we developed a regression model to predict the annotation score of a brain region from the annotation scores, across all attributes, of its connected brain regions. The coefficient of determination (R^2) was used to evaluate the fit of each model. The significance of this fit was computed using a permutation test with 1000 spatial autocorrelation-preserving null permutations and corrected for false discovery rate. In the human connectomes, these permutations consisted in “spun” permutations. In the macaque connectome, these permutations were generated using Moran nulls. To

preserve the dependence between attributes, the same permutation was used across all dependent variables of each model. We also performed dominance analysis on each model, which allowed us to evaluate the contribution of each regressor in the regression models. The significance of these contributions was again computed using a permutation test with spatial autocorrelation-preserving permutations. For the human connectome, the regression models used the E/I ratio, receptor density, T1w/T2w ratio, cortical thickness and gene PC1. For macaque connectomes, the models used the T1w/T2w ratio, cortical thickness and neuron density **(a)** In the structural connectome, we find that the regression model was better at predicting T1w/T2w ratio than regression models built using null annotations. Dominance analysis showed that T1w/T2w ratio and cortical thickness contributed significantly to this performance. **(b)** In the functional connectome, the regression model was better at predicting T1w/T2w ratio, cortical thickness and gene PC1. The dominance analysis shows that these three annotations significantly contributed to the performance of each model **(c)** The regression models built for the macaque connectome performed significantly better than those built using null annotations.

Benjamini, Y., & Yekutieli, D. (2001). The control of the false discovery rate in multiple testing under dependency. *Annals of statistics*, 1165-1188.

Burt, J. B., Demirtaş, M., Eckner, W. J., Navejar, N. M., Ji, J. L., Martin, W. J., ... & Murray, J. D. (2018). Hierarchy of transcriptomic specialization across human cortex captured by structural neuroimaging topography. *Nature neuroscience*, 21(9), 1251-1259.

Huntenburg, J. M., Bazin, P. L., Goulas, A., Tardif, C. L., Villringer, A., & Margulies, D. S. (2017). A systematic relationship between functional connectivity and intracortical myelin in the human cerebral cortex. *Cerebral Cortex*, 27(2), 981-997.

3. Transcription patterns are reduced to the first principal component of >15,000 gene profiles. We are not told what proportion of the variance that captures, but it is likely to be too small to serve as a faithful representation of the underlying signal.

The principal axis (PC1) of transcriptional variation was first studied by Hawrylycz et al. (2012), and has been used in multiple studies since then (see for instance: Burt et al., 2018, Gao et al., 2020; Shafiei et al., 2020, Deco et al., 2021, Dear et al., 2022). It is organized as a gradient from sensorimotor to transmodal regions and is thought to capture a hierarchy of transcriptomic specialization across the human cortex (Huntenburg et al., 2018). It is also related to microcircuit function and neuropsychiatric disorders (Burt et al., 2018). In other words, we chose to study this annotation because it provides valuable information about cortical organization and ensures that our results are comparable with previous literature. We now present this additional information in the methods section:

(“Methods” section, “Human annotations” subsection, paragraph #2):

“The principal axis of transcriptional variation across the human cortex (gene PC1) captures a hierarchy of transcriptomic specialization across the human cortex (Huntenburg et al., 2018, Burt et al., 2018) and has been used in multiple studies (Hawrylycz et al., 2012, Gao et al., 2020; Shafiei et al., 2020, Deco et al., 2021, Dear et al., 2022)”

This gradient captures 20% of the variance in the gene expression matrix, which is consistent with the literature (Burt et al., 2018, Gao et al., 2020, Dear et al., 2022). This information was added to the methods section.

(“Methods” section, “Human annotations” subsection, paragraph #2):

“This principal axis (gene PC1) captures 20% of the gene expression variance. ”

Arnatkevičiūtė, A., Fulcher, B. D., & Fornito, A. (2019). A practical guide to linking brain-wide gene expression and neuroimaging data. *Neuroimage*, 189, 353-367.

Burt, J. B., Demirtaş, M., Eckner, W. J., Navejar, N. M., Ji, J. L., Martin, W. J., ... & Murray, J. D. (2018). Hierarchy of transcriptomic specialization across human cortex captured by structural neuroimaging topography. *Nature neuroscience*, 21(9), 1251-1259.

Dear, R., Seidlitz, J., Markello, R. D., Arnatkeviciute, A., Anderson, K. M., Bethlehem, R. A., ... & Lifespan Brain Chart Consortium. (2022). Three transcriptional axes underpin anatomy, development, and disorders of the human cortex. *bioRxiv*.

Deco, G., Kringelbach, M. L., Arnatkeviciute, A., Oldham, S., Sabaroedin, K., Rogasch, N. C., ... & Fornito, A. (2021). Dynamical consequences of regional heterogeneity in the brain's transcriptional landscape. *Science Advances*, 7(29), eabf4752.

Gao, R., van den Brink, R. L., Pfeffer, T., & Voytek, B. (2020). Neuronal timescales are functionally dynamic and shaped by cortical microarchitecture. *Elife*, 9, e61277.

Hawrylycz, M. J., Lein, E. S., Guillozet-Bongaarts, A. L., Shen, E. H., Ng, L., Miller, J. A., ... & Jones, A. R. (2012). An anatomically comprehensive atlas of the adult human brain transcriptome. *Nature*, 489(7416), 391-399.

Huntenburg, J. M., Bazin, P. L., & Margulies, D. S. (2018). Large-scale gradients in human cortical organization. *Trends in cognitive sciences*, 22(1), 21-31.

Shafiei, G., Markello, R. D., De Wael, R. V., Bernhardt, B. C., Fulcher, B. D., & Misic, B. (2020). Topographic gradients of intrinsic dynamics across neocortex. *Elife*, 9, e62116.

4. Everything rests on a purely linear univariate measure of regional similarity based on Pearson correlation. It is very likely that the diverse annotations studied here exhibit all sorts of non-linear variations, both across the brain and in their associations with each other. For example, the mutual information between two regions may be much higher than a simple linear index suggests. If systematic differences in non-linearity across annotations exist, then the comparison becomes confounded.

To ensure that the interpretation of our results is not biased by the presence of non-linearities in the relationships between the annotations of pairs of connected nodes, we repeated our analyses using a rank-based measure of association that assesses whether two variables are monotonically related and which does not assume a linear relationship between the two variables. These results are shown below, and were added to figures S1, S2 and S3, which present the results of sensitivity and replication experiments for the human structural, human functional and animal connectomes respectively. We also mention these sensitivity analyses in the results, and added additional information about how this measure was computed in the Methods section:

(“Results” section, “Assortativity of cortical attributes” subsection, paragraph #3):

“To ensure that the results are not sensitive to processing choices, we replicated the experiments using different parcellation schemes, single-hemisphere connectomes, an independently acquired dataset, additional spatially-autocorrelation preserving null models and a rank-based assortativity measure (Figs. S1, S2, S3).

(“Methods” section, “Assortativity” subsection, “Ranked-based assortativity” sub-subsection paragraph #1):

“More generally, given an annotation \mathbf{x} , we can define the vectors $\mathbf{x}^{(i)}$ and $\mathbf{x}^{(j)}$ as the annotations of endpoints i and endpoints j across all edges (i,j) in the network. In other words, each entry in the vectors $\mathbf{x}^{(i)}$ and $\mathbf{x}^{(j)}$ represent the annotations of nodes connected by an edge in the network. The assortativity coefficient can then be defined as a weighted Pearson correlation between these two vectors, weighted by the weight of the connection between each edge.

By ranking the annotation scores in the vectors $\mathbf{x}^{(i)}$ and $\mathbf{x}^{(j)}$, we can compute a rank-based assortativity coefficient that corresponds to the weighted Spearman correlation between these two vectors. This rank-based coefficient allows us to evaluate the existence of monotonic relationships between the annotations of connected brain regions.”

Spearman assortativity | For each network, we re-computed the assortativity coefficient of each annotation by relating the annotation scores of pairs of connected nodes using the Spearman rank correlation. The z-assortativity scores for each annotation are shown on the left. Saturated colors indicate significance (FDR corrected). We also re-computed the z-assortativity across thresholded connected connectomes where a given percentile of the shortest connections are removed (right). saturated markers again indicate significance (FDR corrected).

5. What are referred to as spatial nulls are not null models under the assumption of no structure but rather models of local similarity. It is not clear why or how local similarity should be discounted. It is not accidental that the brain exhibits local smoothness and if the objective is to understand variations with distance there is no logic in arbitrarily downweighting proximity, especially when spatial homogeneity may vary substantially across the brain. The

authors use several such presumed nulls, varying the choice across datasets without giving a clear reason.

The spatial autocorrelation-preserving null models used in this experiment implement a specific null hypothesis about the relationship between connectivity, geometry and biological annotations. Namely, that as a result of its spatial embedding, the brain's attributes will, as the Reviewer points out, exhibit local smoothness. Importantly, by using these null models, we do not discount local similarity. On the contrary, the null models disrupt the local similarities between brain regions while preserving the global spatial embedding. As a result, we explicitly evaluate the relationship between the brain's connectivity and these local similarities/spatial heterogeneities.

Had we used spatially-naive permutation models to evaluate the significance of our assortativity results, we would have obtained heavily inflated p-values (Alexander-Bloch et al., 2018; Burt et al., 2020, Markello and Masic, 2021). As proof of concept, we repeated the experiments shown in Figure 3, using spatially-naive permutation models:

As the figure shows, if we had used spatially-naive permutation tests we would have obtained grossly inflated p-values that are asymptotically zero for all annotations. These p-values tell us that brain regions that are connected have similar annotations. This result is however trivial given that most connections are between brain regions that are proximal in space (Bullmore and Sporns, 2012; Roberts et al., 2016; Horvat et al., 2016), and we would obtain similar results with any randomly-generated spatially auto-correlated map. Of course, as the Reviewer points out, these spatial constraints are heterogeneous across the brain, but it is specifically by applying null models that preserve the global spatial embedding of the brain that we are able to evaluate the role of these heterogeneous spatial constraints on the relationship between the brain's connectivity and micro-architectural attributes.

Regarding the specific choice of null models for each dataset, we prioritized the use of spatial permutation nulls (i.e. spin tests) since they tend to be more conservative and reduce the risk of obtaining false positives, compared to the parameterized null models (Markello and Masic, 2021). Permutation nulls require spherical projections of the brain, which are available for the human fsaverage5 cortical mesh in FreeSurfer. As a result, human datasets were evaluated against permutation nulls. However, spherical projections are not available for the animal datasets. We therefore primarily used parameterized nulls (Moran nulls) for these datasets. For completeness, we replicated all results with the Burt nulls and replicated the human results with the Moran nulls, i.e.

- human data: spin nulls, Burt nulls and Moran nulls
- animal data: Burt nulls and Moran nulls

We have clarified this in the revised methods section.

("Methods" section, "Spatial autocorrelation-preserving null annotations" subsection, paragraph #1):

We controlled for the brain's spatial constraints using null models that preserve the spatial autocorrelation of the empirical attributes. The use of spatial permutation nulls (spin nulls) were prioritized since they tend to be more conservative (Markello & Music, 2021). These nulls require the use of spherical projections of the brain, which were not available in animal datasets. For the animal datasets, we therefore relied on a parameterized null model that uses Moran spectral randomization (Moran nulls) (vos de Wael et al., 2020). All the results were also replicated with a third null model originally proposed by Burt and colleagues (Burt nulls), and the results obtained with spin nulls (i.e. for the human connectomes) were also replicated with the Moran nulls. The spin, Moran and Burt nulls were respectively implemented with the neuromaps (<https://github.com/netneurolab/neuromaps>) (Markello et al., 2022), brainspace (<https://github.com/MICA-MNI/BrainSpace>) and brainSMASH (<https://github.com/murraylab/brainsmash>) (Burt et al., 2020) toolboxes.

Alexander-Bloch, A. F., Shou, H., Liu, S., Satterthwaite, T. D., Glahn, D. C., Shinohara, R. T., ... & Raznahan, A. (2018). On testing for spatial correspondence between maps of human brain structure and function. *Neuroimage*, 178, 540-551.

Bullmore, E., & Sporns, O. (2012). The economy of brain network organization. *Nature reviews neuroscience*, 13(5), 336-349.

Burt, J. B., Helmer, M., Shinn, M., Anticevic, A., & Murray, J. D. (2020). Generative modeling of brain maps with spatial autocorrelation. *NeuroImage*, 220, 117038.

Horvát, S., Gămănuț, R., Ercsey-Ravasz, M., Magrou, L., Gămănuț, B., Van Essen, D. C., ... & Kennedy, H. (2016). Spatial embedding and wiring cost constrain the functional layout of the cortical network of rodents and primates. *PLoS biology*, 14(7), e1002512.

Markello, R. D., & Misic, B. (2021). Comparing spatial null models for brain maps. *NeuroImage*, 236, 118052.

Roberts, J. A., Perry, A., Lord, A. R., Roberts, G., Mitchell, P. B., Smith, R. E., ... & Breakspear, M. (2016). The contribution of geometry to the human connectome. *Neuroimage*, 124, 379-393.

Vos de Wael, R., Benkarim, O., Paquola, C., Lariviere, S., Royer, J., Tavakol, S., ... & Bernhardt, B. C. (2020). BrainSpace: a toolbox for the analysis of macroscale gradients in neuroimaging and connectomics datasets. *Communications biology*, 3(1), 1-10.

6. Though we are not given the number of tests the authors clearly perform many, yet there is no mention of any correction for multiple comparisons. This is not trivial where correlation between tests is likely, as in the regional comparisons. In any event, without it there is little we can conclude from the p values cited.

We now correct for multiple comparisons across annotations. More specifically, we correct for False discovery rate (FDR) using the Benjamini-Yekutieli procedure (Benjamini & Yekutieli, 2001), which

controls for false discovery rate under arbitrary (both positive and negative) dependence assumptions. The corrections did not change the main conclusions drawn for the “Assortativity of cortical attributes” and “Geometric contributions to assortativity” sections. We updated the p-values in the revised manuscript:

(“Results” section, “Assortativity of cortical attributes” subsection, paragraph #2):

“We find that annotations are not significantly assortative on the human structural connectome, while gene PC1 ($z\text{-assort}= 3.15$, $p_{\text{spin}}=0.0013$, $p_{\text{FDR}}=0.0049$), T1w/T2w ($z\text{-assort}= 6.02$, $p_{\text{spin}}<0.0001$, $p_{\text{FDR}}<0.0001$) and cortical thickness ($z\text{-assort}=3.63$, $p_{\text{spin}}=0.0003$, $p_{\text{FDR}}=0.0017$) are significantly assortative on the human functional connectome. In the macaque connectome, we observe a significant difference between the assortativity of T1w/T2w and null annotations ($z\text{-assort}= 3.95$, $p_{\text{morán}}=0.0008$, $p_{\text{FDR}}=0.002$) as well as between neuron density and null annotations ($z\text{-assort}= 3.97$, $p_{\text{morán}}<0.0001$, $p_{\text{FDR}}<0.0001$). No significant difference is observed for the cortical thickness. In the mouse connectome, no significant difference is observed for gene PC1”

(“Results” section, “Geometric contributions to assortativity” subsection, paragraph #2):

“We find that as short-distance connections are removed – leaving behind the longest connections – the standardized assortativity of all annotations across all four connectomes decreases (Fig.4). Notably, with 80% of the human structural connectome’s connections removed, four annotations become significantly disassortative: E/I ratio ($z\text{-assort}= -2.94$, $p_{\text{spin}}=0.0037$, $p_{\text{FDR}}=0.021$), gene PC1 ($z\text{-assort}= -2.88$, $p_{\text{spin}}=0.0028$, $p_{\text{FDR}}=0.021$), T1w/T2w ratio ($z\text{-assort}= -2.27$, $p_{\text{spin}}=0.018$, $p_{\text{FDR}}=0.049$) and cortical thickness ($z\text{-assort}=-2.49$, $p_{\text{spin}}=0.01$, $p_{\text{FDR}}=0.039$)”

For the “Heterophilic mixing of cortical attributes” section, we find that the mixing patterns between the laminar thickness of layer VI and the laminar thicknesses of layers III, V and VI are not significantly assortative in the functional connectome with FDR correction. This however does not affect our main finding, which we now clarify in our reply to Reviewer #2: that these mixing patterns highlight a relationship between the topographic distribution of laminar thicknesses and the brain’s functional hierarchy, defined as the principal axis of variance in the functional connectome. The assortative relationship between VaChT and NAT is still significant after FDR correction. We updated the p-values in the revised manuscript:

(“Results” section, “Heterophilic mixing of cortical attributes” subsection, paragraph #4):

“We also find a significant assortative relationship in the functional connectome between vesicular acetylcholine transporters (VaChT) and NAT ($z\text{-assort}= 5.10$, $p_{\text{spin}}=0.0003$, $p_{\text{FDR}}=0.028$).

For the “Local assortative mixing” section, our main finding was that words associated with executive functions (e.g. working memory, intelligence, etc.) were positively associated with the homophilic ratio while words associated with learning and memory were negatively associated with the homophilic ratio.

We ran additional experiments to confirm that these associations are indeed significant. Using the Cognitive Atlas (Poldrack et al., 2011) we grouped each of the 123 terms into 11 cognitive categories. These categories consist of “Action”, “Learning and Memory”, “Emotion”, “Attention”, “Reasoning and

Decision Making”, “Executive/Cognitive control”, “Social Function”, “Perception”, “Motivation”, “Language” and “other”. Lists of terms associated with each category can be found here: <http://www.cognitiveatlas.org/concepts/categories/all>. We then computed the average correlations with the homophilic ratio for all the brain maps within each category and computed the significance of the average correlations using a permutation test. This permutation test consisted in randomly permuting the categories associated with each term and computing the average correlations obtained from these permuted categories. This was repeated 10000 times and a p-value was computed by comparing the average correlations obtained with the empirical categories with the null distribution of average correlations. With this test, we find, significantly negative averaged correlations for terms associated with executive/cognitive control ($r=-0.21$, $p_{FDR}<0.0001$) and action ($r=-0.20$, $p_{FDR}=0.022$) and significantly positive correlations for terms associated with learning and memory ($r=0.11$, $p_{FDR}<0.0001$) and emotion ($r=0.15$, $p_{FDR}=0.009$). The methodology is now described in the “Methods” section, and the results are now presented in the “Results” section:

(“Results section, “Local assortative mixing” subsection, paragraph #4):

We then averaged the correlations obtained for cognitive terms associated with 11 different cognitive categories (Fig. 6e). We find that, on average, activation maps of cognitive terms associated to the “Action” and “Executive/Cognitive control” categories are negatively correlated with the average homophilic ratio brain map (action: $r=-0.20$, $p_{spin}=0.0026$, $p_{FDR}=0.022$; executive/cognitive control: $r=-0.21$, $p_{spin}<0.0001$, $p_{FDR}<0.0001$). Conversely, we find that cognitive terms associated to the “learning and memory” and “emotion” categories are, on average, positively correlated with the average homophilic ratio brain map (learning and memory: $r=0.11$, $p_{spin}=0.0004$, $p_{FDR}=0.007$; emotion: $r=0.15$, $p_{spin}=0.0008$, $p_{FDR}=0.009$).

Benjamini, Y., & Yekutieli, D. (2001). The control of the false discovery rate in multiple testing under dependency. *Annals of statistics*, 1165-1188.

Poldrack, R. A., Kittur, A., Kalar, D., Miller, E., Seppa, C., Gil, Y., ... & Bilder, R. M. (2011). The cognitive atlas: toward a knowledge foundation for cognitive neuroscience. *Frontiers in neuroinformatics*, 5, 17.

Reviewer #2:

The authors add an important analysis to the current studies of brain connectomes – they investigate how regional biological attributes affect white matter connectivity, thus linking traditional neuroanatomic approaches with current brain connectivity studies. Specifically, the authors study assortative mixing in annotated connectomes, hypothesizing that regions with similar vs. dissimilar micro-architectonic attributes (such as molecular, cellular and laminar annotations, including gene expression, neurotransmitter receptors, neuron density, laminar thickness and intracortical myelin) show distinct connectivity patterns. The analyses are based on four connectome datasets from three species, i.e. humans, macaques and mice and the authors use a novel null model to assess the contribution of spatial constraints (i.e. to control for spatial autocorrelation). They observe that regions with similar annotations tend to connect with each other and that micro-architecturally diverse regions are connected via long-distance projections.

These are very interesting data and analyses. My main questions are regarding the interpretation of results.

The results regarding heterophilic mixing of brain areas with a prominent layer 4 are particularly interesting. The authors write that “layer IV is most prominent in the primary visual cortex, while the thicknesses of layers III, V and VI increase along the sensory processing hierarchy” Can this fully explain the strong effects observed? Are the authors referring to visual sensory processing, or sensory processing in general (I assume the former)? The authors could (rather easily?) test their hypothesis of a gradual change of heterophilic mixing of primary visual cortex along the visual processing hierarchy.

These are important questions. In the sentence cited, we refer to the visual processing hierarchy: we cite the work of Wagstyl et al. (2020), who showed relationships between the laminar thicknesses of each region and their physical position relative to V1. Importantly, we did not hypothesize that there is a gradual change of heterophilic mixing of the primary visual cortex along the visual processing hierarchy. Rather, we highlighted this result from the literature because we hypothesized it could reflect, more generally, a topographic organization of laminar thicknesses from sensory to association cortex.

We made this hypothesis because of the relationship between assortativity and the principal axis of variance of the brain’s functional connectivity matrix (FC-PC1), which is often described as a functional hierarchy ranging from unimodal to transmodal association brain regions (Huntenburg et al., 2018). By definition, brain regions that have similar scores on the FC-PC1 axis are densely interconnected. Thus, if an annotation is strongly correlated with this axis, then densely interconnected brain regions also have similar annotation scores and the network is, as a result, assortative.

We have revised the manuscript to explicitly state and contextualize this result: that laminar thickness varies along the unimodal-transmodal hierarchy, explaining the patterns of assortativity observed between the different layers.

(“Results” section, “Heterophilic mixing of cortical attributes” subsection, paragraph #3):

“Several salient associations emerge that are consistent with prior intuitions and qualitative descriptions in the literature (Fig. 5b). For the laminar thickness, we find consistent mixing patterns for both the structural and functional connectomes. Namely, layers III, V and VI are assortative with respect to each other, but disassortative with respect to layer IV. In other words, we find that brain areas with a prominent layer IV tend to preferentially connect with brain areas with thin layers III, V and VI, whereas brain regions with prominent layers III, V and VI tend to preferentially connect with each other. Interestingly, in the functional connectome, these mixing patterns can be explained by the relationship between each laminar thickness map and the functional hierarchy, defined as the main axis of variance in the brain’s functional connectivity matrix (Margulies et al., 2016, Huntenburg et al., 2018). Indeed, assortative mixing between two annotations arises when attributes have positive or negative relationships with the main axis of brain connectivity, while disassortative mixing arises when attributes have opposite relationships with this axis (Fig. S6ab). For the patterns of assortativity observed between layer IV and layers III, V and VI, it has been shown that the thickness of layer IV is most prominent in the primary visual cortex (Wagstyl et al., 2020), while the thicknesses of layers III, V and VI increases along the visual processing hierarchy (Wagstyl et al., 2020). Similarly, we find here that the thickness of layer IV is positively correlated with the functional hierarchy while the thicknesses of layers III, V and VI are negatively correlated with the functional hierarchy (Fig. S6c). These opposing relationships with the functional hierarchy, such that a prominent layer IV is associated with unimodal brain regions while prominent layer III, V and VI are associated with multimodal regions explain the mixing patterns observed (Fig. S6c).”

(“Methods” section, “Functional hierarchy” subsection):

To further clarify these ideas, we added a new supplementary figure:

Figure S6. Relationship between the principal axis of connectivity and assortativity | (a) The principal axis of connectivity in the human functional connectome (FC PC1) is described as a functional hierarchy ranging from unimodal to transmodal cortex (Margulies et al., 2016, Huntenburg et al., 2018). It can be computed by applying principal component analysis on the functional connectivity matrix. **(b)** Brain regions are ordered along FC PC1 based on their connectivity profile: densely interconnected regions are grouped together on this axis (left). When a pair of annotations is correlated with FC PC1, then densely interconnected regions have similar annotation scores (e.g. score highly on two annotations **X** and **Y**). As a result, this pair of annotations will be assortative on the functional connectome. **(c)** We computed the Pearson correlation between the laminar thickness brain maps and FC PC1 and show that layers II and IV are positively correlated with FC PC1 while layers VI, V, III and I are negatively correlated with FC PC1 (left). For each pair of laminar thickness maps, we then evaluated whether they had similar or opposing relationships with FC PC1 by computing the product of their correlations with FC PC1 (middle). We then correlated the products with the z-assortativity of each pair of annotations and found a significant relationship between the two ($r=0.75$, $p=0.00008$). In other words, we show that the assortativity between pairs of laminar thickness maps can be explained by their relationship with the functional hierarchy. **(d)** We also evaluated the correlation between each receptor density map and FC PC1, and again compared the product of the

correlations with the z-assortativity of each pair of annotations. We again also find a significant relationship ($r=0.74$; $p<10^{-34}$).

Huntenburg, J. M., Bazin, P. L., & Margulies, D. S. (2018). Large-scale gradients in human cortical organization. *Trends in cognitive sciences*, 22(1), 21-31.

Margulies, D. S., Ghosh, S. S., Goulas, A., Falkiewicz, M., Huntenburg, J. M., Langs, G., ... & Smallwood, J. (2016). Situating the default-mode network along a principal gradient of macroscale cortical organization. *Proceedings of the National Academy of Sciences*, 113(44), 12574-12579.

Wagstyl, K., Larocque, S., Cucurull, G., Lepage, C., Cohen, J. P., Bludau, S., ... & Evans, A. C. (2020). BigBrain 3D atlas of cortical layers: cortical and laminar thickness gradients diverge in sensory and motor cortices. *PLoS biology*, 18(4), e3000678.

The authors write that they „find evidence of disassortative mixing for pairs of receptors that are predominantly expressed in brain regions on opposite ends of the processing hierarchy“ - could they explain this in more detail, i.e. what hierarchy and how this was defined and what receptor subtypes are involved? Indeed, most receptors and their subtypes are involved in a multitude of very different functions.

Here we are again referring to the unimodal-transmodal hierarchy, defined as the principal axis of variance of the brain’s functional connectivity matrix. The new text in the Results of the revised manuscript - shown in our response to the Reviewer’s previous comment above - explicitly defines the term “hierarchy” and clarifies its relationship with assortative mixing.

Regarding the receptor subtypes involved, we now highlight the significant relationships in Figure 5 of our updated manuscript. We find 10 pairs of receptors and transporters that are significantly disassortative. These pairs are now presented in Table S1:

Annotations		z-assortativity	p (spin)	p (FDR)
5HTT	5HT2a	-4.68	<0.0001	<0.0001
5HTT	5HT4	-4.23	<0.0001	<0.0001
5HTT	CB1	-3.16	0.0008	0.049
5HTT	MU	-3.67	0.0001	0.012
NAT	MU	-3.28	<0.0001	<0.0001
NAT	5HT2a	-7.81	<0.0001	<0.0001
NAT	5HT4	-5.84	<0.0001	<0.0001
NAT	M1	-4.25	0.0002	0.020
NAT	mGluR5	-3.54	0.0006	0.044
VAcHT	5HT2a	-4.63	<0.0001	<0.0001

Table S1: Significantly disassortative mixing between receptors and transporters in FC

A clear pattern emerges, whereby each pair involves a transporter (5HTT/NAT/VAcHT) and a receptor. Again, the assortativity between a pair of annotations is directly related to the extent to which their distribution overlaps with the functional hierarchy: transporter maps tend to be anticorrelated with the unimodal-transmodal hierarchy, while the identified receptor maps tend to be positively correlated with the unimodal-transmodal hierarchy. We show these correlations in Fig S6, introduced in our response to the Reviewer’s previous comment above. Altogether, these results highlight a network-mediated balance between transporter and receptor density.

We now clarify the pairs of receptors and transporters identified by the analysis:

(“Results” section, “Heterophilic mixing of cortical attributes” subsection, paragraph #4):

“This general idea also extends to receptors where we broadly find evidence of disassortative mixing for pairs of receptors and transporters predominantly expressed in brain regions on opposite ends of this functional hierarchy. More specifically, we find 10 pairs of receptors and transporters that are significantly disassortative (Table S1). A clear pattern emerges, whereby each pair involves a transporter (5HTT, NAT or VAcHT) and a receptor. This is in line with their relationship with the functional hierarchy: transporter maps tend to be anticorrelated with the unimodal-transmodal hierarchy, while the identified receptor maps tend to be positively correlated with the unimodal-transmodal hierarchy (Fig. S6d). Altogether, these results highlight a network-mediated balance between transporter and receptor density.”

How were receptors selected for these analyses? There are 6 different subtypes of serotonin receptors, but no ionotropic glutamate receptors and only one metabotropic glutamate receptor; specifically, why are NMDAR and AMPA receptors missing, given that glutamate is the main excitatory neurotransmitter?

The receptors selected for these analyses are those for which radioligands exist and for which whole brain density maps generated using positron emission tomography (PET) are publicly available. These brain maps were compiled by our group in a publicly available dataset of 18 receptors and transporters from 9 different neurotransmitter systems (Hansen et al., 2022)

At the time of writing we did not have access to a high-quality PET image for any of the ionotropic glutamate receptors. However, we recently added to this publicly available atlas an NMDA receptor density brain map that was acquired by Galovic and colleagues (2021, medRxiv; 2021, NeuroImage). We incorporated this NMDA receptor density brain map into the analyses presented in this revised manuscript.

Galovic, M., Al-Diwani, A., Vivekananda, U., Torrealdea, F., Erlandsson, K., Fryer, T. D., ... & Koepp, M. J. (2021). In vivo NMDA receptor function in people with NMDA receptor antibody encephalitis. *medRxiv*.

Galovic, M., Erlandsson, K., Fryer, T. D., Hong, Y. T., Manavaki, R., Sari, H., ... & Koepp, M. J. (2021). Validation of a combined image derived input function and venous sampling approach for the quantification of [18F] GE-179 PET binding in the brain. *NeuroImage*, 237, 118194.

Hansen, J. Y., Shafiei, G., Markello, R. D., Smart, K., Cox, S. M., Nørgaard, M., ... & Misic, B. (2022). Mapping neurotransmitter systems to the structural and functional organization of the human neocortex. *Nature neuroscience*, 1-13.

I am also not fully convinced yet about the interpretation of how the “homophilic ratio of a brain region shapes its functional specialization”. The homophilic ratio was positively correlated with memory performance, and it appears plausible to interpret that e.g. the medial temporal lobe is “involved in integrating signals from multiple specialized circuits”. However, the homophilic ratio was negative correlated with executive function – but executive functions specifically also require integration of signals from very different brain regions and circuits, and they are not only subserved by a highly interconnected module centering around the DLPFC.

We agree that it is surprising that a highly integrative region like DLPFC tends to have more homophilic relationships (with respect to biological interactions) than other brain regions. However,

our finding that brain regions associated with executive functions tend to connect to other regions with similar micro-architectural properties does not necessarily imply that they do not have connectivity profiles that support integrative functions.

We ran additional analyses to explore this idea. We clustered the structural connectome into 9 communities of highly interconnected regions and computed the mean homophilic ratio and mean node strength inside each community. This analysis confirmed that regions in DLPFC have a small homophilic ratio (i.e. are homophilic with respect to biological annotations), but also showed that DLPFC is the community with the largest mean nodal strength. In other words, DLPFC is highly connected to the rest of the network, consistent with the fact that it is highly integrated, but it preferentially connects to areas with very similar biological annotations.

(“Results” section, “Local assortative mixing” subsection, paragraph #3):

“We find a significant relationship for both mean connection distance ($r=0.35$, $p_{\text{spin}} < 0.0001$), and node strength ($r=0.21$, $p_{\text{spin}}=0.021$). In other words, disassortative regions have, on average, longer connections, which is consistent with our previous findings that long distance connections tend to be disassortative (Fig. 4). Our results highlight a general trend in the structural connectome whereby brain regions that have large node strength (i.e. hub regions) tend to be more disassortative. In other words, the hub regions of the brain tend to connect to regions that have dissimilar micro-architectural attributes.

We next explored whether these findings are consistent across communities of the brain. We clustered the human structural connectomes into 9 communities of highly interconnected regions and computed the mean homophilic ratio and mean node strength inside each community. We find that the relationships between node strength, mean connection distance and homophilic ratio hold for all communities, with the exception of the dorsolateral prefrontal community (Fig. S10). Brain regions in this community have, on average, the largest node strengths but the smallest homophilic ratios. In other words, contrary to the other hub regions of the brain, nodes in DLPFC tend to preferentially connect to regions that have similar micro-architectural attributes.”

a | communities of the structural connectome

b | homophilic, geometric and topological properties

Figure S10. Homophilic ratios in the communities of the structural connectome | (a) We clustered the human structural connectome into 9 communities of highly interconnected brain regions and evaluated the relationship between the homophilic, geometric and topological properties of each community. (b) We computed the mean homophilic ratio, node strength and mean connection distance in each community and evaluated how these averaged scores relate to each other. We find positive relationships between node strength and homophilic ratio (left), as well as between mean connection distance and homophilic ratio (right). The dorsolateral prefrontal cortex, however, is an outlier: it has the largest mean node strengths and mean connection distances, but the smallest homophilic ratios.

Furthermore, to avoid causal claims, we rephrase our interpretation that “homophilic ratio of a brain region shapes its functional specialization” to “homophilic ratio of a brain region is associated with functional specialization”

The relevant methodology describing the community detection algorithm was added to the methods section:

(“Methods” section, “Community detection” subsection):

“Communities are groups of nodes with dense connectivity among each other. The Louvain method was used to identify a community assignment or partition that maximizes the quality function Q (Blondel et al., 2008):

$$Q = \frac{1}{2m} \sum_{ij} [A_{ij} - \gamma \frac{s_i s_j}{2m}] \delta(c_i, c_j)$$

where A_{ij} is the weight of connection between nodes i and j , s_i and s_j are the directed strengths of i and j , m is a normalizing constant, c_i is the community assignment of node i and the Kronecker δ -function $\delta(u,v)$ is defined as 1 if $u = v$ and 0 otherwise. The resolution parameter γ scales the importance of the null model and effectively controls the size of the detected communities: larger communities are more likely to be detected when $\gamma < 1$ and smaller communities (with fewer nodes in each community) are more likely to be detected when $\gamma > 1$.

To detect stable community assignments for the structural connectome we initiated the algorithm 100 times at each value of the resolution parameter and consensus clustering was used to identify the most representative partitions (Lancichinetti & Fortunato, 2012). This procedure was repeated for a range of 100 resolutions between $\gamma = 0.25$ and $\gamma = 7.5$. We then quantified the similarity between pairs of consensus partitions using the z score of the Rand index (Red et al., 2011). Ultimately, we chose the 9-communities consensus partition obtained at $\gamma = 1.23$ because the generated partitions obtained for this value of γ showed high mutual similarity and persisted through stretches of γ values. The whole procedure was implemented using code available in the netneurotools python toolbox (<https://github.com/netneurolab/netneurotools>).”

Blondel, V. D., Guillaume, J. L., Lambiotte, R., & Lefebvre, E. (2008). Fast unfolding of communities in large networks. *Journal of statistical mechanics: theory and experiment*, 2008(10), P10008.

Lancichinetti, A., & Fortunato, S. (2012). Consensus clustering in complex networks. *Scientific reports*, 2(1), 1-7.

Red, V., Kelsic, E. D., Mucha, P. J., & Porter, M. A. (2011). Comparing community structure to characteristics in online collegiate social networks. *SIAM review*, 53(3), 526-543.

It would also be good to see the correlation coefficients for all 123 cognitive and behavioural terms to get a better idea about the distribution of effects (might be best in the suppl. Material)

We agree that it would be good to show the correlation coefficients for all 123 terms. We now include a supplementary figure showing these correlation coefficients, and refer to this figure in the results section.

(“Results” section, “Local assortative mixing” subsection, paragraph #4):

“Correlations for individual cognitive terms are shown in Fig. S11.”

Figure S11. Neurosynth correlations | Correlation coefficients between the averaged homophilic ratios of micro-architectural attributes and 123 brain maps associated with cognitive and behavioural terms. Each map represents the probabilistic association between the term and individual voxel activation. We find 59 negative relationships (top) and 64 positive relationships (bottom). Asterisks and colored bars denote the significant relationships ($p_{\text{spin}} < 0.05$).

REVIEWERS' COMMENTS

Reviewer #2 (Remarks to the Author):

Many thanks to the authors for their clarifications and additional analyses - great job and this is an excellent paper.